# Script: Graph-Structured and Query-Conditioned Semantic Token Pruning for Multimodal Large Language Models

**Zhongyu Yang**[*]                                                          *zy4028@hw.ac.uk*
*BCML, Heriot-Watt University*
**Dannong Xu**[*]                                                      *danielxu0208@gmail.com*
*BCML, Heriot-Watt University*
**Wei Pang**                                                               *w.pang@hw.ac.uk*
*BCML, Heriot-Watt University*
**Yingfang Yuan**[†]                                                        *y.yuan@hw.ac.uk*
*BCML, Heriot-Watt University*

**Reviewed on OpenReview:** *https://openreview.net/forum?id=F6xKzbgcHq*

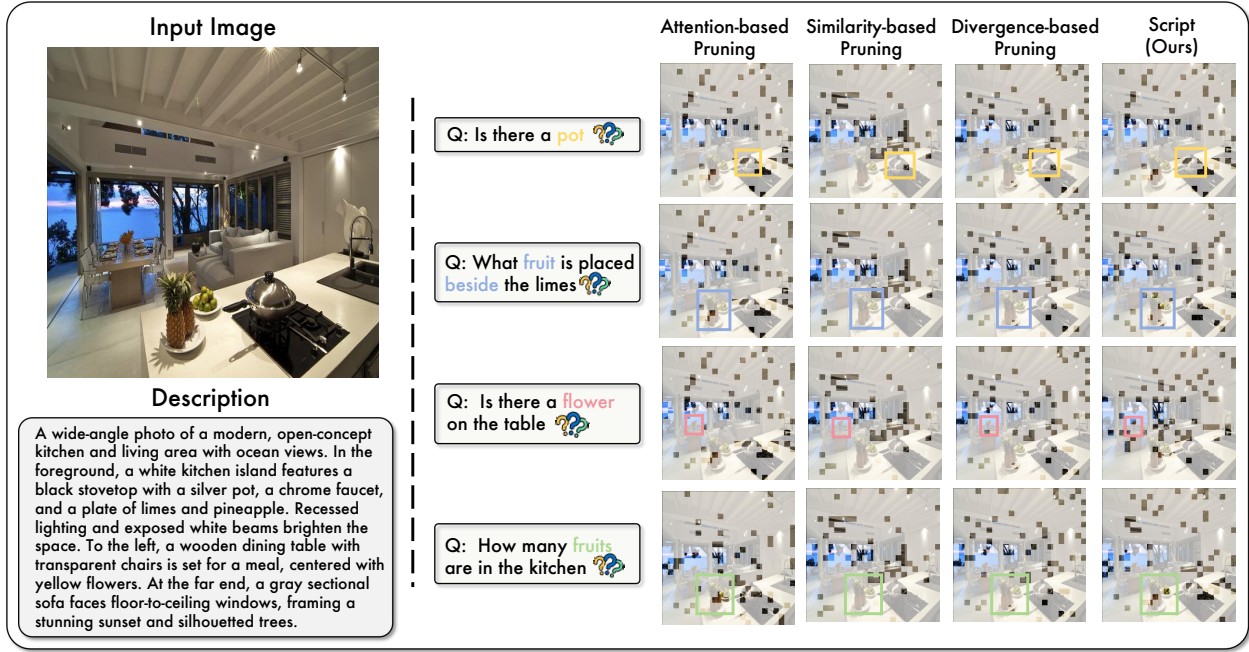

Figure 1: **Comparison of different token pruning methods.** Attention-based and similarity-based methods prune tokens using attention scores and similarity scores, respectively. In contrast, divergence-based methods detect changes in model performance and retain tokens that cause minimal impact. **Script** (Graph-**S**tructured and Que**R**y-**C**ond**I**tioned **T**oken **P**runing) combines graph-structured reduction of visual redundancy and query-conditioned semantic token selection to enable efficient pruning in MLLMs. In this example, Script successfully preserves key visual cues, such as the **silver pot** on the stove, the **pineapple** beside the **limes**, and the **flowers** on the table. Other methods fail to retain consistently.

## Abstract

The rapid growth of visual tokens in multimodal large language models (MLLMs) leads to excessive memory consumption and inference latency, especially when handling high-resolution images and videos. Token pruning is a technique used to mitigate this issue by removing redundancy, but existing methods often ignore relevance to the user query or suffer from the

---

[1][*] Equal contribution
[2][†] Correspond Author

limitations of attention mechanisms, reducing their adaptability and effectiveness. To address these challenges, we propose Script, a plug-and-play pruning method that requires no retraining and generalizes across diverse MLLMs. Script comprises two modules: a graph-structured pruning module that removes visually redundant tokens, and a query-conditioned semantic pruning module that preserves query-relevant visual information. Together, they enhance performance on multimodal tasks. Experiments on fourteen benchmarks across image and video understanding tasks show that Script consistently achieves higher model efficiency and predictive accuracy compared to existing pruning methods. On LLaVA-NeXT-7B, it achieves up to 6.8× prefill speedup and 10× FLOP reduction, while retaining 96.88% of the original performance.

## 1 Introduction

Recent advances in large language models (LLMs) (Touvron et al., 2023; Bai et al., 2023; Yang et al., 2024; 2025a) have substantially enhanced language understanding and reasoning, forming the backbone of general-purpose multimodal systems, including vision-language models and embodied agents operating in real-world scenarios. Building on this progress, multimodal large language models (MLLMs) (Li et al., 2024a; Chen et al., 2024d;c; Zhang et al., 2025) extend LLMs by integrating vision encoders, enabling joint reasoning over both visual and textual modalities. This integration empowers MLLMs to excel at diverse vision-language tasks such as visual question answering and image captioning.

However, deploying MLLMs in practical scenarios such as mobile agents or interactive assistants is constrained by the high computational cost of handling high-resolution or temporally extended visual inputs. Patch-based vision encoders typically tokenize each visual input into hundreds or even thousands of tokens (Luo et al., 2024; Guo et al., 2024; Zhang et al., 2023; Chen et al., 2024b; Li et al., 2025b; Liu et al., 2025; Bai et al., 2025). In contrast to compact textual inputs, visual representations often contain hundreds or thousands of tokens, which increases memory usage and inference latency. The explosion of vision tokens, compounded by the quadratic complexity of attention mechanisms (Vaswani et al., 2017), creates a significant bottleneck for latency- and memory-sensitive scenarios such as mobile deployment, online inference, and edge-based vision-language applications. To mitigate this inefficiency, visual token pruning has emerged as a promising strategy. An effective pruning method should minimize computational cost while preserving task performance within resource-constrained environments.

Existing visual token pruning methods can be broadly categorized into three main paradigms: (1) attention-based methods, which retain tokens with high model-assigned importance, commonly referred to as attention scores (Chen et al., 2024a; Xing et al., 2025; Zhang et al., 2024c); (2) similarity-based methods, which identify and eliminate redundant visual tokens based on feature similarity (Bolya et al., 2023; Zhang et al., 2024b; Wen et al., 2025b; Jeddi et al., 2025); and (3) divergence-minimization methods, which prune tokens by minimizing the change in the model's output (Alvar et al., 2025; Ye et al., 2025).

However, existing pruning methods face two fundamental challenges in query-conditioned scenarios. **First**, attention-based approaches depend on raw attention scores, which are susceptible to the *attention sink* issue (Barbero et al., 2025), often overlooking critical tokens. Moreover, assigning similar scores to adjacent or semantically similar tokens can reduce pruning efficiency (Yang et al., 2025b). As shown in Figure 1, such methods fail to preserve the token representing the flower, despite its clear relevance to the query. **Second**, similarity- and divergence-based methods lack explicit query conditioning (Alvar et al., 2025; Li et al., 2025a). Although they address visual

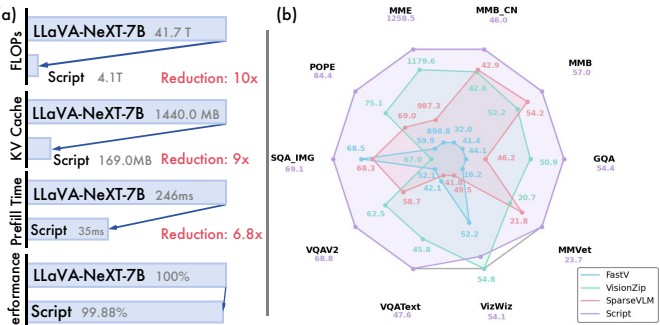

Figure 2: (a) Efficiency Analysis on LLaVA-NeXT-7B under 88.9% reduction. (b) Comparison with other baselines on LLaVA-1.5-7B under 94.4% reduction.

redundancy or output stability, they generate fixed token subsets regardless of the input query, which can lead to the omission of query-referenced objects, such as the pineapple and lime shown in Figure 1.

To address these limitations, we propose **Script**, a training-free and architecture-agnostic token pruning approach that combines graph-structured pruning (GSP) and query-conditioned semantic pruning (QCSP). On one hand, we build a bipartite graph to structure visual token redundancy, effectively identifying redundant tokens with lower computational cost. On the other hand, to eliminate reliance on attention scores and avoid issues such as attention sink, Script explicitly models interactions between the query and visual tokens, identifying query-relevant tokens using Determinantal Point Processes (DPP), a probabilistic model that favors diverse and semantically meaningful subsets. Together, these two modules enable effective token pruning by jointly considering visual redundancy and input query relevance. As illustrated in Figure 2, Script trims 88.9% tokens yet preserves 99.88% accuracy on POPE, and consistently outperforms other baselines on 10 benchmarks. In summary, our contributions are as follows:

- We identify the core limitations of existing pruning paradigms in query-conditioned scenarios, highlighting their inability to adaptively preserve task-relevant content.

- We propose Script, a training-free pruning method that combines graph-based redundancy reduction with query-aware token selection via DPP.

- Experiments across fourteen benchmarks demonstrate that Script consistently overperforms other baselines, and achieves up to **10×** speedup while retaining **96.88%** of original performance.

## 2 Related Work

**Scalability Challenges in MLLMs.** MLLMs combine visual and textual modalities to support general-purpose perception and reasoning (Alayrac et al., 2022; Zhu et al., 2024; Chen et al., 2023; Yu et al., 2024; Liu et al., 2024b; Cai et al., 2024). However, visual inputs introduce substantial token overhead due to spatial redundancy and high dimensionality, exacerbating the quadratic complexity of Transformer attention. For example, LLaVA-1.5 (Liu et al., 2023) generates 576 tokens from a $336 \times 336$ image, which is already 4–5× longer than typical text-only prompts. High-resolution models like LLaVA-NeXT (Liu et al., 2024a) and video models like LongVA (Zhang et al., 2024a) push this further, producing tens or even hundreds of thousands of tokens per input, leading to untenable memory and compute demands for many downstream applications. Consequently, such models often encounter memory bottlenecks, latency spikes, or even inference failures, especially under real-time or edge deployment scenarios (Papa et al., 2023; Li et al., 2025c), where computational resources are inherently limited. These challenges have motivated a growing body of work on token pruning, aiming to reduce input size while preserving task-relevant semantics and ideally without retraining or architectural changes.

**Token Pruning Strategies.** Efforts to mitigate visual token overhead in MLLMs can be broadly categorized into four approaches, each with distinct assumptions and limitations. (1) Pre-fusion compression, which downsamples or selects tokens before vision-language fusion (Li et al., 2024b; Hu et al., 2024), often requires model retraining or structural modifications, hindering plug-and-play usage. (2) Attention-based pruning, which selects tokens using attention scores (Chen et al., 2024a; Ye et al., 2025), suffers from attention drift (Wen et al., 2025a), misguiding token importance, and limiting compatibility with optimized backends such as FlashAttention (Dao, 2024). (3) Pre-language fusion pruning, which drops tokens before language-level alignment (Shang et al., 2024; Song et al., 2025), is often tightly coupled to specific vision backbones, limiting transferability across architectures. (4) Similarity-based pruning, which eliminates redundant tokens using intra-image feature similarity (Wen et al., 2025b; Alvar et al., 2025), is generally efficient and model-agnostic, but often ignores query semantics, leading to the loss of task-relevant content.

While existing approaches partially alleviate visual redundancy, few simultaneously achieve the trifecta of query relevance, architectural generality, and training-free deployment. In contrast, our proposed **Script** is explicitly query-aware, visual diversity, entirely model-agnostic, and operates without any additional training.

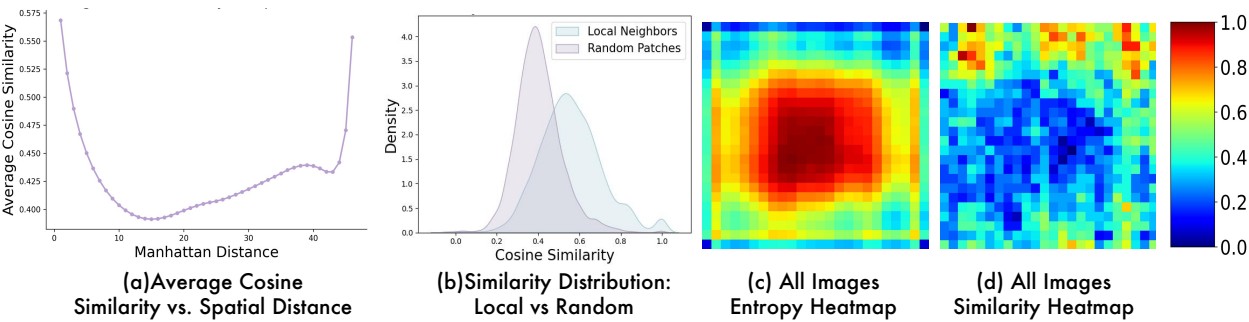

Figure 3: **Token redundancy visualized via similarity and entropy on 10,000 COCO images.**

## 3 Preliminary

In this section, within the context of the task Text Visual Question Answering (TextVQA), we empirically investigate two key aspects of visual tokens in MLLMs: visual redundancy and query-conditioned relevance. In addition, we present a theoretical formulation of efficient query-conditioned token selection. The insights from both empirical and theoretical analysis inform the design of our **Script**.

### 3.1 Research Objective

Existing MLLMs (Liu et al., 2023; 2025; Xu et al., 2025) typically comprise three components: a vision encoder $f_v$, a multimodal projector $g$, and a large language model $f_\phi$. After $f_v$ processes an input image $\boldsymbol{X}_v$, the projector $g$ maps the features into a visual-token sequence $\boldsymbol{H}_v \in \mathbb{R}^{n \times d}$, where $n$ is the token count and $d$ is the token dimension. The language model $f_\phi$ processes the visual tokens $\boldsymbol{H}_v$ and the tokenized query $\boldsymbol{H}_q$ to generate an answer grounded in the image. Typically, $n \gg |\boldsymbol{H}_q|$, i.e., the visual sequence is much longer than the textual query. Equation 1 defines vision-token pruning by seeking the subset $\tilde{\boldsymbol{H}}_v^*$ that preserves model performance while using the fewest tokens.

$$\tilde{\boldsymbol{H}}_v^* = \arg \min_{\tilde{\boldsymbol{H}}_v \subseteq \boldsymbol{H}_v, \, |\tilde{\boldsymbol{H}}_v|=m} \mathcal{L}\left(f_\phi([\tilde{\boldsymbol{H}}_v; \boldsymbol{H}_q]), f_\phi([\boldsymbol{H}_v; \boldsymbol{H}_q])\right). \tag{1}$$

Here, $\mathcal{L}$ measures the discrepancy between model outputs before and after pruning, serving as a proxy for performance loss. The candidate subset $\tilde{\boldsymbol{H}}_v \subset \boldsymbol{H}_v$ contains $m$ tokens, with $m < n$. To ensure query-aware pruning, it is essential to explicitly capture the semantic interactions between visual tokens $\boldsymbol{H}_v$ and their corresponding query tokens $\boldsymbol{H}_q$. These query-specific interactions are crucial for identifying the most query-relevant subset $\tilde{\boldsymbol{H}}_v$. At the same time, assessing the visual redundancy between $\tilde{\boldsymbol{H}}_v$ and the full set $\boldsymbol{H}_v$ is vital for promoting diversity in token selection, which is important for generalizing across a wide range of tasks. Together, these two aspects contribute to forming an informative and compact token subset.

### 3.2 Rethinking Visual Redundancy

In natural images, spatially adjacent tokens often encode highly overlapping content, resulting in what we refer to as local redundancy. Carefully identifying and removing such redundancy can significantly reduce computation without compromising accuracy.

To investigate local redundancy, we adopt the approach from (Wang et al., 2024) and compute Manhattan distances and cosine similarities between vision-token pairs. We group similarity values by distance and calculate the average for each group. As shown in Figure 4(a), cosine similarity is highest for nearby tokens and decreases until a

Table 1: Comparison of redundancy estimation methods with a 90% pruning ratio on the POPE Benchmark. **Sim. (%)**: Percentage of similarity pairs considered. **Time**: Inference time in milliseconds. **IOU@K**: Intersection-over-Union with top-K selections. **Acc.**: Classification accuracy (%). **Speedup**: Relative time speedup compared to Full Graph.

| Method | Sim. (%) | Time | IOU@K | Acc. | Speedup |
|---|---|---|---|---|---|
| Exhaustive | 100 | 100 | 1.00 | 85.14 | 1.00x |
| Bipartite | ∼50 | 35 | 0.93 | 84.49 | 2.81x |
| Random | 0 | 34 | 0.52 | 78.63 | 2.82x |

distance of roughly 15. Beyond this, we observe that similarity increases again at longer distances, indicating that long-range token redundancy also exists and deserves attention. Furthermore, we visualize the similarity distributions for neighboring tokens ($hop = 1$) and randomly sampled neighbor token pairs ($hop > 1$). As shown in Figure 4(b), the number of neighboring tokens with similarity greater than 0.5 is consistently higher than that of random token pairs. These results indicate that neighboring (i.e., local) tokens encode more similar visual features, but long-range redundancy should also be taken into account, as it clearly exists.

> **Insight 1: Redundancy Exists Beyond Local Neighborhoods**
>
> Spatially adjacent tokens show high similarity (local redundancy), yet long-range pairs still share appreciable similarity, revealing long-range redundancy. Effective pruning must therefore model both local and long-range redundancy to avoid information loss.

To further validate local redundancy and its estimation via cosine similarity, we introduce local information entropy as a distribution-level measure of neighborhood compactness and compare it with cosine similarity. For each token, we extract its $3 \times 3$ Moore neighbourhood $\mathcal{N}(v) = \{\mathbf{h}_i \mid \|\text{pos}(i) - \text{pos}(j)\|_1 \leq 1\}$ ($hop$=1). We then project all vectors in the neighborhood onto their first principal component and discretize the resulting scalar values into 20 equal-width bins to estimate the probability distribution $\{p_i\}$. The local information entropy is defined as:

$$\mathcal{H}_v = -\sum_{i=1}^{n} p_i \log p_i, \tag{2}$$

where $p_i = p_i + \epsilon$ and $\epsilon = 10^{-8}$. A compact neighborhood typically exhibits low $\mathcal{H}_v$, as its token representations are similar and concentrated in a few bins. In contrast, a diverse neighborhood naturally yields higher $\mathcal{H}_v$, reflecting greater variation among neighboring tokens. As shown in Figures 3(c) and (d), regions with low local similarity typically exhibit high entropy, confirming local redundancy and validating cosine similarity as an effective proxy.

> **Insight 2: Cosine Similarity is a Valid Redundancy Proxy**
>
> Local information entropy inversely correlates with cosine similarity. Low-entropy (predictable) regions coincide with high neighbor similarity.

Motivated by the validated role of cosine similarity in capturing visual redundancy and the need for computational efficiency, we design a lightweight bipartite graph-based estimator that approximates both local and long-range redundancy using cosine similarity alone. This estimator forms the core of the graph-structured pruning module detailed in Section 4.1.

> **Insight 3: Lightweight Bipartite Graph Yields Scalable Redundancy Estimation**
>
> By weighting edges with cosine similarity, the even–odd bipartite graph achieves redundancy estimation comparable to exhaustive computation, but with significantly lower computational cost.

Tokens are represented as nodes, divided into even- and odd-indexed sets; each node in one set connects to all nodes in the other, with edge weights defined by cosine similarity. These weights enable efficient redundancy estimation and guide token removal. As shown in Table 1, empirical results confirm the effectiveness of the bipartite graph-based approach. On the POPE Benchmark with a 90% pruning ratio, it achieves an accuracy of 84.49%, which is close to that of the exhaustive approach (85.14%), while being nearly three times faster. Meanwhile, it retains 93% IOU agreement with the exhaustive approach, indicating that most key tokens are preserved. In contrast, the random approach is equally fast but drops to 78.63% accuracy and only 0.52 IOU, failing to preserve semantically relevant tokens.

### 3.3 Rethinking Query Relevance

Recent studies (Zhang et al., 2024b; Yang et al., 2025b) address visual token redundancy using text-agnostic pruning strategies that retain tokens with high [CLS] attention scores from the vision encoder's final layer. However, these methods often fail to incorporate the query explicitly. They are also limited by inherent issues in attention mechanisms, such as the attention sink effect, where a disproportionate amount of attention is assigned to the first token in a sequence, regardless of its semantic relevance. As a result, these methods tend to preserve tokens from visually salient or high-attention regions. Meanwhile, they may discard less prominent areas, such as backgrounds or textures, which can still be crucial to answering the query.

This observation raises a critical question: *Is output-layer [CLS] attention sufficient to capture all query-relevant visual information?* Empirical observations suggest such attention-based strategies often overemphasize dominant foreground objects while neglecting background elements relevant to the query. As shown in Figure 1, attention focuses on the *pot* and *fruit* in the foreground but overlooks the actual target, the *flower* in the background. This results in incorrect outputs. Moreover, both divergence-based and similarity-based approaches also fail to adapt to the user query in this example.

> **Insight 4: CLS Attention Misses Query-Relevant Background Tokens**
>
> Attention-based relevance measures tend to focus on visually salient foregrounds and can miss query-relevant background tokens, underscoring the need for explicitly query-aware relevance scoring.

### 3.4 Rethinking Query-Relevant Visual Token Selection

Let the query tokens be $\mathbf{H}_q = \{\mathbf{h}_j^{(q)}\}$, and define their mean embedding as:

$$\mathbf{h}_\mu^{(q)} = \frac{1}{\ell} \sum_{j=1}^{\ell} \mathbf{h}_j^{(q)}. \tag{3}$$

The relevance score $q_i$ of each visual token $\mathbf{h}_i$ is computed as the cosine similarity between $\mathbf{h}_i$ and the mean query embedding $\mathbf{h}_\mu^{(q)}$:

$$r_i = \frac{\mathbf{h}_i^\top \mathbf{h}_\mu^{(q)}}{\|\mathbf{h}_i\| \|\mathbf{h}_\mu^{(q)}\|}, \quad \text{for } i = 1, \ldots, n. \tag{4}$$

We construct a diagonal matrix $\boldsymbol{Q} = \text{diag}(r_1, \ldots, r_n) \in \mathbb{R}^{n \times n}$ to encode the query relevance of each visual token. To encourage diversity among selected tokens, we define a similarity matrix $\boldsymbol{S}^{(v)} \in \mathbb{R}^{n \times n}$, where $\boldsymbol{S}_{ij}^{(v)}$ measures the pairwise similarity between vision tokens $\mathbf{h}_i$ and $\mathbf{h}_j$. We assume $\boldsymbol{S}^{(v)}$ is symmetric and positive semidefinite, which approximately holds when using cosine similarity among normalized features. This formulation naturally aligns with modeling token selection as a $k$-DPP (Determinantal Point Process), which promotes subsets that balance individual relevance with collective diversity. We therefore construct the following $k$-DPP kernel (note that $\boldsymbol{L}$ is a positive-semidefinite kernel, **not** a loss function) to integrate both query relevance and feature diversity:

$$\boldsymbol{L} = \boldsymbol{Q}^{1/2} \boldsymbol{S}^{(v)} \boldsymbol{Q}^{1/2}, \tag{5}$$

where $\boldsymbol{Q}$ explicitly encodes query alignment and $\boldsymbol{S}^{(v)}$ impliciltly encourages feature diversity.

> **Insight 5: $\mathbf{S}^{(v)}$ Promotes Diversity**
>
> For any subset $\mathcal{I}$, $\det(\boldsymbol{L}_\mathcal{I})$ equals the squared volume of the parallelotope spanned by the selected token vectors (see Appendix B). If $\boldsymbol{S}^{(v)}$ were the identity matrix, this volume reduces to $\prod_{i \in \mathcal{I}} q_i$, i.e., relevance only. Introducing off-diagonal similarities lowers the determinant when two tokens are similar, thus maximizing $\det(\boldsymbol{L}_\mathcal{I})$ encourages mutually orthogonal (diverse) token subsets.

A $k$-DPP assigns to each subset $\mathcal{I} \subseteq [n]$ of fixed size $k$ a probability proportional to $\det(\boldsymbol{L}_{\mathcal{I}})$, the determinant of the principal submatrix of $\boldsymbol{L}$ indexed by $\mathcal{I}$. This determinant reflects the volume spanned by the selected vectors, and is larger when the selected tokens are both informative and mutually distinct. Expanding the determinant yields:

$$
\begin{aligned}
\det(\boldsymbol{L}_{\mathcal{I}}) &= \det(\boldsymbol{Q}_{\mathcal{I}}^{1/2} \boldsymbol{S}_{\mathcal{I}}^{(v)} \boldsymbol{Q}_{\mathcal{I}}^{1/2}) \\
&= \det(\boldsymbol{Q}_{\mathcal{I}}^{1/2})^2 \cdot \det(\boldsymbol{S}_{\mathcal{I}}^{(v)}) \\
&= \left( \prod_{i \in \mathcal{I}} q_i \right) \cdot \det(\boldsymbol{S}_{\mathcal{I}}^{(v)}).
\end{aligned}
\tag{6}
$$

This decomposition highlights the trade-off between query relevance (via $\prod_i q_i$) and visual diversity (via $\det(\boldsymbol{S}_{\mathcal{I}}^{(v)})$). Based on this, we define the final surrogate objective for selecting an informative token subset:

$$
\mathcal{I}^* = \arg \max_{\mathcal{I} \subseteq [n], |\mathcal{I}| = m} \prod_{i \in \mathcal{I}} q_i \cdot \det(\boldsymbol{S}_{\mathcal{I}}^{(v)}).
\tag{7}
$$

> **Insight 6: $k$-DPP Balances Relevance and Diversity via Geometry**
>
> The $k$-DPP determinant can be analogized to the feature-space volume spanned by selected tokens, thus optimizing it jointly maximizes query relevance and inter-token diversity in a single objective.

## 4 Method

Motivated by the empirical analysis in Section 3, we introduce Script, as illustrated in Figure 4. Script progressively refines token selection through two main modules: (1) Graph-Structured Pruning (GSP), which removes visually redundant tokens using a bipartite similarity graph; and (2) Query-Conditioned Semantic Pruning (QCSP), which consists of two steps: Query-Conditioned Relevance Scoring (QCRS), computing the semantic relevance of each token with respect to the input query, and Diversity-Preserving Selection via DPP, selecting a compact, diverse subset of relevant tokens. The outputs from GSP and QCSP are then intersected to obtain the final token subset $\tilde{\boldsymbol{H}}_v^*$.

### 4.1 Graph-Structured Pruning

Transformer-based visual encoders often generate dense token sequences with substantial visual redundancy across both local and long-range contexts, as identified in Section 3.2. To mitigate this, we propose an inference-time pruning strategy based on bipartite similarity graphs without relying on model parameters.

**Bipartite Graph Construction.** Given visual token embeddings $\boldsymbol{H}_v = [\mathbf{h}_1, \ldots, \mathbf{h}_n] \in \mathbb{R}^{n \times d}$, we construct a bipartite graph by partitioning tokens into two disjoint node sets $V_{\text{src}}$ and $V_{\text{dst}}$ via alternating index assignment (e.g., even indices to $V_{\text{src}}$, odd-indexed to $V_{\text{dst}}$) (Buchholz, 2024). The bipartite graph is defined as $G = (V_{\text{src}}, V_{\text{dst}}, \boldsymbol{S}^{(v)})$, where each node in $V_{\text{src}}$ is fully connected to every node in $V_{\text{dst}}$, and the edge weight $\boldsymbol{S}_{ij}^{(v)} \in \boldsymbol{S}^{(v)}$ represents the cosine similarity between tokens $t_i$ and $t_j$:

$$
\boldsymbol{S}_{ij}^{(v)} = \frac{\bar{\mathbf{h}}_i^{\text{src}} \cdot \bar{\mathbf{h}}_j^{\text{dst}}}{\|\bar{\mathbf{h}}_i^{\text{src}}\| \|\bar{\mathbf{h}}_j^{\text{dst}}\|}.
\tag{8}
$$

In this way, $G$ explicitly structures token redundancy, with similarity scores providing insights into both local and global redundancy. Notably, this bipartite graph design reduces computational cost by up to 75% compared to conventional similarity-based pruning methods. As empirically validated in Section 3.2, most redundancy lies in local neighborhoods and is thus well-captured by the bipartite approximation effectively. Overall, $V_{\text{src}}$ and $V_{\text{dst}}$ jointly offer a structured, low-cost representation of $\boldsymbol{H}_v$ for redundancy modeling.

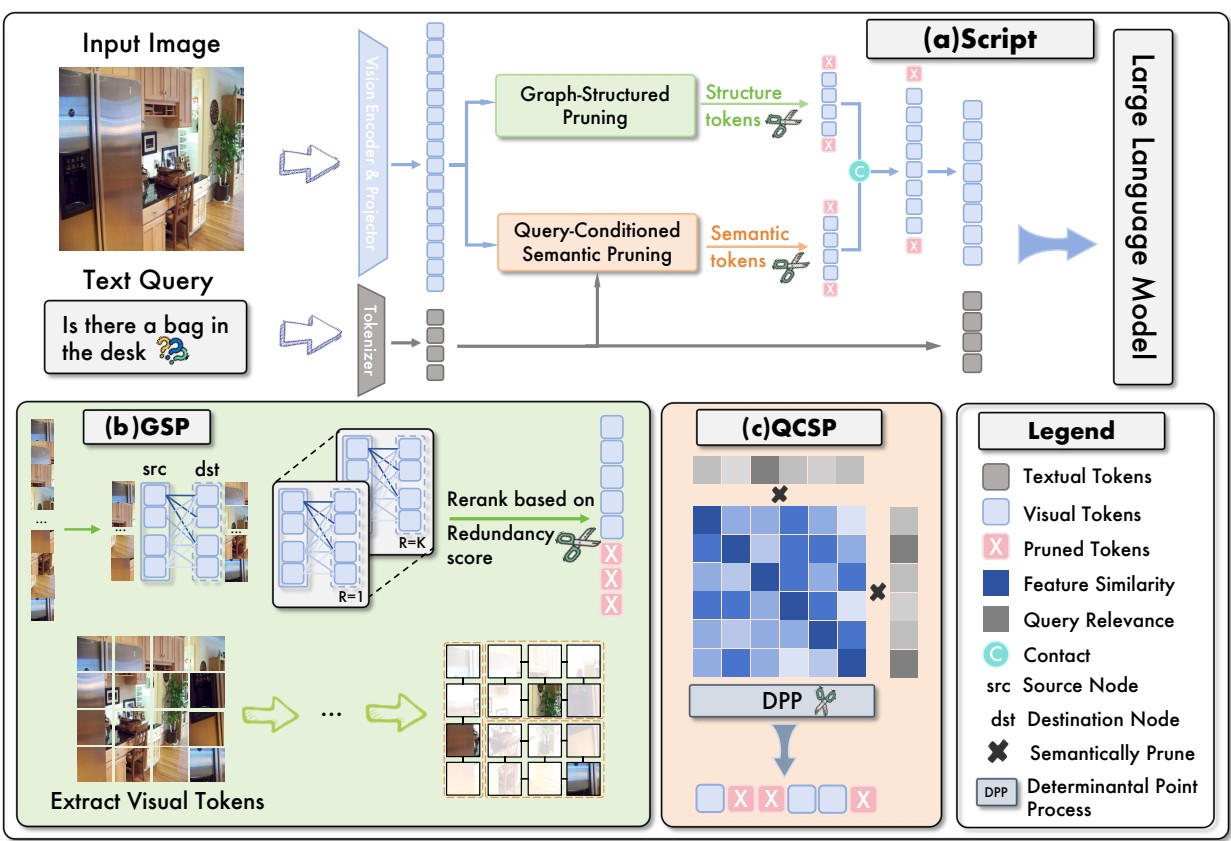

Figure 4: **Overview of Script**, a three-stage pruning framework: (a) overall architecture; (b) Query-Conditioned Semantic Pruning (QCSP); (c) Graph-Structured Pruning (GSP). Together, these modules remove semantically irrelevant and visually redundant tokens through a joint selection process.

**Redundancy Scoring.** To identify redundant tokens, we retain all similarity edges above a threshold $\tau$, forming a pruned subgraph $G_\tau$, where each node is connected to highly similar neighbors. For a token $t_i \in V_{\mathrm{src}}$, we define its degree as: $d(t_i) = \sum_{j \in V_{\mathrm{dst}}} \mathbb{I}\{S_{ij}^{(v)} \geq \tau\}$, where $\mathbb{I}$ denotes the indicator function, returning 1 if the condition is true and 0 otherwise. Nodes with high degrees in $G_\tau$ are identified as structurally redundant, typically indicating that tokens represent repeated visual content in either local or long-range contexts. To further distinguish tokens with similar degrees, we use the average similarity $\mu(t_i)$ between token $t_i$ and its neighbors in $G_\tau$, which captures their typical similarity. For cases where $t_i$ is an isolated node in $G_\tau$, we define a fallback similarity $\tilde{\mu}(t_i)$ based on the mean similarity between $t_i$ and all tokens in $V_{\mathrm{dst}}$ or $V_{\mathrm{src}}$ under the full graph $G$. The final redundancy score is then defined as:

$$\mathrm{score}(t_i) = \begin{cases} d(t_i) \cdot \exp\left(\gamma(\mu(t_i) - \tau)\right), & d(t_i) > 0 \text{ in } G_\tau, \\ \tilde{\mu}(t_i) \text{ in } G, & \text{otherwise,} \end{cases} \tag{9}$$

where $\gamma$ is a tunable scaling factor. The score, computed using $d(t_i)$ and $\mu(t_i)$ with exponential weighting, is designed to improve ranking contrast between highly and moderately redundant tokens.

**Pruning Rate and Token Removal.** Based on the computed redundancy scores, we rank all visual tokens in descending order and discard the top $p$ most redundant ones, where $p$ denotes the pruning ratio. This yields a structurally pruned candidate set $\tilde{H}_v$, which is then combined with the token set semantically aligned with the query, produced in the next stage, to jointly form $\tilde{H}_v^*$.

Table 2: **Performance comparisons on LLaVA-1.5-7B (Liu et al., 2023) across 10 image understanding benchmarks**. The best results in each setting are **bolded**, and the second-best are underlined.

| Method | Venue | GQA | MMB | MMB$^{CN}$ | MME | POPE | SQA$^{IMG}$ | VQA$^{V2}$ | VQA$^{Text}$ | VizWiz | MMVet | Acc. | Relative |
|---|---|---|---|---|---|---|---|---|---|---|---|---|---|
| *Upper Bound, 576 Tokens (100%), 3.817 TFLOPs* | | | | | | | | | | | | | |
| LLaVA-1.5-7B (Liu et al., 2023) | *Nips'23* | 61.94 | 64.09 | 58.10 | 1507.06 | 86.96 | 69.41 | 78.50 | 58.20 | 50.32 | 31.82 | 63.47 | 100% |
| *Retain 192 Tokens in Average (↓ 66.7%), ~1.253 TFLOPs* | | | | | | | | | | | | | |
| FastV (Chen et al., 2024a) | *ECCV'24* | 58.09 | 63.40 | 54.04 | 1490.65 | 82.30 | 68.96 | 72.88 | 54.09 | 54.97 | 29.59 | 61.29 | 96.56% |
| TRIM (Song et al., 2025) | *COLING'25* | 58.18 | 59.19 | 51.02 | 1405.29 | 78.09 | 67.87 | 74.86 | 54.29 | 48.23 | 30.82 | 59.28 | 93.41% |
| VisionZip (Yang et al., 2025b) | *CVPR'25* | 59.24 | 63.75 | 53.52 | 1445.67 | 86.63 | 68.37 | 75.44 | 54.89 | 53.97 | 30.74 | 59.28 | 93.40% |
| DivPrune (Alvar et al., 2025) | *CVPR'25* | 59.91 | 62.54 | 43.00 | 1436.39 | **87.56** | 69.31 | 75.10 | 53.38 | **55.33** | 30.96 | 60.89 | 95.94% |
| SparseVLM (Zhang et al., 2024c) | *ICML'25* | 59.54 | 63.46 | 55.20 | 1450.07 | 86.49 | 68.67 | 75.61 | 56.93 | 51.23 | 31.24 | 62.08 | 97.82% |
| **Script (Ours)** | *Proposed* | **60.82** | **63.85** | **57.55** | **1493.87** | 87.42 | **69.49** | **77.62** | **57.81** | 53.80 | **31.40** | **63.45** | **100.00%** |
| *Retain 128 Tokens in Average (↓ 77.8%), ~0.833 TFLOPs* | | | | | | | | | | | | | |
| FastV (Chen et al., 2024a) | *ECCV'24* | 56.85 | 62.46 | 53.61 | **1440.19** | 80.24 | 68.67 | 71.68 | 52.21 | 55.51 | 28.07 | 60.13 | 94.74% |
| TRIM (Song et al., 2025) | *COLING'25* | 56.85 | 58.42 | 48.66 | 1359.86 | 77.95 | 68.47 | 74.42 | 53.74 | 47.62 | 29.35 | 58.35 | 91.93% |
| VisionZip (Yang et al., 2025b) | *CVPR'25* | 57.87 | 62.20 | 53.18 | 1429.75 | 85.20 | 68.22 | 74.62 | 54.12 | 54.39 | 27.39 | 60.87 | 95.90% |
| DivPrune (Alvar et al., 2025) | *CVPR'25* | 59.16 | 61.86 | 42.16 | 1412.97 | **87.36** | **69.01** | 74.19 | 51.92 | **56.21** | 28.30 | 60.08 | 94.66% |
| SparseVLM (Zhang et al., 2024c) | *ICML'25* | 58.41 | 62.75 | 53.48 | 1429.41 | 86.22 | 68.62 | 73.86 | 56.29 | 51.83 | 29.12 | 61.21 | 96.43% |
| **Script (Ours)** | *Proposed* | **60.27** | **63.10** | **55.85** | 1431.24 | 87.32 | 68.79 | **76.55** | 56.45 | 53.97 | **31.09** | **62.51** | **98.49%** |
| *Retain 64 Tokens in Average (↓ 88.9%), ~0.415 TFLOPs* | | | | | | | | | | | | | |
| FastV (Chen et al., 2024a) | *ECCV'24* | 53.59 | 59.62 | 50.08 | 1366.33 | 75.34 | 68.11 | 58.71 | 51.62 | 54.74 | 27.06 | 56.72 | 89.36% |
| TRIM (Song et al., 2025) | *COLING'25* | 56.35 | 55.68 | 46.39 | 1288.31 | 77.80 | 68.22 | 73.10 | 51.69 | 48.29 | 28.29 | 56.58 | 89.14% |
| VisionZip (Yang et al., 2025b) | *CVPR'25* | 55.47 | 60.82 | 51.29 | 1374.67 | 81.68 | 68.96 | 71.59 | 52.72 | 54.81 | 27.13 | 59.32 | 93.46% |
| DivPrune (Alvar et al., 2025) | *CVPR'25* | 57.74 | 59.28 | 39.20 | 1368.28 | 86.51 | 68.22 | 72.35 | 54.51 | 57.55 | 27.39 | 59.12 | 93.14% |
| SparseVLM (Zhang et al., 2024c) | *ICML'25* | 53.80 | 60.14 | 50.68 | 1295.16 | 80.92 | 68.46 | 68.24 | 53.67 | 50.11 | 24.92 | 57.67 | 90.86% |
| **Script (Ours)** | *Proposed* | **59.28** | **61.90** | **52.93** | **1412.08** | **86.95** | 68.65 | **75.08** | **55.20** | 54.31 | **29.96** | **61.49** | **96.88%** |
| *Retain 32 Tokens in Average (↓ 94.5%), ~0.208 TFLOPs* | | | | | | | | | | | | | |
| FastV (Chen et al., 2024a) | *ECCV'24* | 49.61 | 54.64 | 43.99 | 1108.35 | 68.59 | 68.26 | 56.48 | 49.85 | 54.22 | 24.50 | 52.56 | 82.80% |
| TRIM (Song et al., 2025) | *COLING'25* | 54.35 | 53.31 | 43.27 | 1215.42 | 77.54 | 68.22 | 69.54 | 49.12 | 38.41 | 25.92 | 54.05 | 85.15% |
| VisionZip (Yang et al., 2025b) | *CVPR'25* | 53.32 | 58.85 | 49.31 | 1306.97 | 78.18 | 68.62 | 67.51 | 50.76 | 55.47 | 24.95 | 57.23 | 90.17% |
| DivPrune (Alvar et al., 2025) | *CVPR'25* | 55.11 | 58.93 | 35.99 | 1303.26 | 83.93 | 68.57 | 69.41 | 51.46 | 57.26 | 25.73 | 57.16 | 90.05% |
| SparseVLM (Zhang et al., 2024c) | *ICML'25* | 51.74 | 59.64 | 48.64 | 1181.05 | 77.27 | 69.35 | 63.25 | 51.28 | 49.62 | 23.38 | 55.32 | 87.16% |
| **Script (Ours)** | *Proposed* | **57.58** | **61.28** | **50.38** | **1338.27** | **86.77** | 68.75 | **73.25** | **53.10** | 53.77 | **27.57** | **59.93** | **94.42%** |
| *Retain 16 Tokens in Average (↓ 97.3%), ~0.103 TFLOPs* | | | | | | | | | | | | | |
| FastV (Chen et al., 2024a) | *ECCV'24* | 44.08 | 41.41 | 32.04 | 898.83 | 59.89 | 68.51 | 52.06 | 42.15 | 52.16 | 16.24 | 45.35 | 71.45% |
| TRIM (Song et al., 2025) | *COLING'25* | 50.81 | 45.28 | 38.46 | 1085.66 | 81.10 | 66.98 | 61.48 | 39.80 | 35.26 | 20.49 | 49.39 | 77.82% |
| VisionZip (Yang et al., 2025b) | *CVPR'25* | 50.88 | 52.23 | 42.61 | 1179.56 | 75.11 | 66.98 | 62.48 | 45.77 | 54.78 | 20.69 | 53.05 | 83.58% |
| DivPrune (Alvar et al., 2025) | *CVPR'25* | 51.10 | 53.09 | 31.76 | 1235.71 | 75.98 | 69.21 | 63.89 | 41.15 | 55.04 | 19.86 | 52.29 | 82.38% |
| SparseVLM (Zhang et al., 2024c) | *ICML'25* | 46.16 | 54.16 | 42.93 | 987.32 | 69.00 | 68.26 | 58.72 | 41.78 | 49.47 | 21.84 | 50.17 | 79.04% |
| **Script (Ours)** | *Proposed* | **54.42** | **57.02** | **45.98** | **1258.55** | **84.44** | 69.12 | **68.77** | **47.65** | 54.09 | **23.71** | **56.81** | **89.51%** |

## 4.2 Query-Conditioned Semantic Pruning

While the GSP module eliminates visually redundant tokens, it does not guarantee semantic alignment with the user query. To address this limitation, we introduce the QCSP module, which comprises QCRS for each token and diversity-preserving selection via DPP based on the relevance scores.

**Query-Conditioned Relevance Scoring.** We begin by computing an average query embedding $\mathbf{h}_\mu^{(q)}$ via mean pooling over $\boldsymbol{H}_q$. Each visual token $\mathbf{h}_i$ is then scored by cosine similarity to $\mathbf{h}_\mu^{(q)}$, yielding a raw query relevance vector $\mathbf{r}_{\text{raw}} = [r_1 \dots r_n]$ (see Eq. 4). To ensure consistency across samples, we apply min-max normalization to obtain $\mathbf{r}_{\text{norm}} \in [0,1]^n$.

**Query-Conditioned Kernel Construction.** To support diversity-preserving selection of query-relevant tokens, we construct a query-aware DPP kernel that integrates token relevance with visual similarity. Importantly, visual similarity in this context helps refine query-relevant token selection rather than explicitly promote visual diversity, which is addressed by the GSP module.

Specifically, let $\boldsymbol{S}'^{(v)} \in \mathbb{R}^{n \times n}$ be the cosine similarity matrix of $\ell_2$-normalized visual token embeddings:

$$\boldsymbol{S}'^{(v)}_{ij} = \left\langle \frac{\mathbf{h}_i}{\|\mathbf{h}_i\|}, \frac{\mathbf{h}_j}{\|\mathbf{h}_j\|} \right\rangle. \tag{10}$$

Then, we define a query-conditioned kernel $\tilde{\boldsymbol{L}} \in \mathbb{R}^{n \times n}$ by reweighting the pairwise visual similarity matrix $\boldsymbol{S}'^{(v)}$ with normalized token-to-query relevance scores $\mathbf{r}_{\text{norm}}$: $\tilde{\boldsymbol{L}} = \text{diag}(\mathbf{r}_{\text{norm}}) \cdot \boldsymbol{S}'^{(v)} \cdot \text{diag}(\mathbf{r}_{\text{norm}})$. Intuitively, $\tilde{\boldsymbol{L}}$ captures diversity within token relevance, promoting token selections that are both query-aligned and diverse. Its symmetric and positive semi-definite structure makes it well-suited for $k$-DPP-based selection [1].

---

[1] $k$ is not a tunable hyperparameter but a direct control of the number of retained tokens. It is conceptually equivalent to the pruning ratio $p$ in GSP, with the correspondence $k = (1-p)n$, where $n$ is the number of original tokens.

Table 3: **Performance comparisons on** LLaVA-NeXT-7B **(Liu et al., 2024a) across 9 image under-standing benchmarks**. The best results in each setting are **bolded**, and the second-best are underlined.

| Method | Venue | GQA | MMB | MMB$^{\text{CN}}$ | MME | POPE | SQA$^{\text{IMG}}$ | VQA$^{\text{V2}}$ | VQA$^{\text{Text}}$ | VizWiz | MMVet | Acc. | Relative |
|---|---|---|---|---|---|---|---|---|---|---|---|---|---|
| *Upper Bound, 2,880 Tokens (100%), ~20.825 TFLOPs* | | | | | | | | | | | | | |
| LLaVA-NeXT-7B (Liu et al., 2024a) | *CVPR'24* | 62.83 | 65.81 | 57.65 | 1504.72 | 86.92 | 67.59 | 81.20 | 60.38 | 55.65 | 41.11 | 65.43 | 100% |
| *Retain 640 Tokens in Average (↓ 77.8%), ~ 4.627 TFLOPs* | | | | | | | | | | | | | |
| FastV (Chen et al., 2024a) | *ECCV'24* | 58.93 | 63.14 | 53.88 | 1412.86 | 79.96 | 67.32 | 77.06 | 58.15 | 53.94 | 39.17 | 62.22 | 95.09% |
| TRIM (Song et al., 2025) | *COLING'25* | 61.13 | 65.83 | 55.79 | 1473.62 | 86.92 | 66.71 | 78.36 | 54.85 | 54.12 | 37.16 | 63.46 | 96.98% |
| VisionZip (Yang et al., 2025b) | *CVPR'25* | 60.84 | 64.52 | 56.28 | 1408.67 | 85.86 | 66.36 | 79.13 | 59.90 | 55.46 | 38.79 | 63.76 | 97.44% |
| DivPrune (Alvar et al., 2025) | *CVPR'25* | 60.58 | 65.77 | 57.31 | 1457.48 | 86.69 | 66.76 | 79.26 | 57.04 | 55.62 | 38.16 | 64.01 | 97.82% |
| SparseVLM (Zhang et al., 2024c) | *ICML'25* | 60.32 | 65.66 | 56.81 | 1426.83 | 85.98 | 67.27 | 77.11 | 59.64 | 55.58 | 39.48 | 63.92 | 97.69% |
| **Script (Ours)** | *Proposed* | **62.64** | **65.98** | **57.76** | **1481.98** | **87.65** | 67.43 | 80.47 | 58.92 | **55.71** | **41.90** | **65.35** | **99.88%** |
| *Retain 320 Tokens in Average (↓ 88.9%), ~2.314 TFLOPs* | | | | | | | | | | | | | |
| FastV (Chen et al., 2024a) | *ECCV'24* | 52.17 | 53.87 | 44.51 | 1178.61 | 72.54 | 65.67 | 66.52 | 52.30 | 51.33 | 26.58 | 54.44 | 83.21% |
| TRIM (Song et al., 2025) | *COLING'25* | 59.94 | 63.58 | 51.01 | 1426.49 | 86.17 | 66.22 | 74.27 | 51.02 | 53.94 | 32.77 | 61.02 | 93.27% |
| VisionZip (Yang et al., 2025b) | *CVPR'25* | 59.11 | 62.89 | 53.39 | 1360.61 | 84.44 | 65.71 | 76.41 | **58.53** | 55.27 | 36.25 | 62.00 | 94.76% |
| DivPrune (Alvar et al., 2025) | *CVPR'25* | 59.24 | 64.03 | 55.58 | 1418.46 | 84.92 | 67.11 | 77.24 | 56.17 | 54.97 | 35.72 | 62.59 | 95.66% |
| SparseVLM (Zhang et al., 2024c) | *ICML'25* | 58.75 | 64.18 | 54.86 | 1399.77 | 85.16 | 66.14 | 75.62 | 56.55 | 55.22 | 37.94 | 62.44 | 95.43% |
| **Script (Ours)** | *Proposed* | **61.36** | **65.39** | **55.82** | **1452.85** | **87.22** | **67.80** | **78.45** | 57.24 | **55.51** | **38.12** | **63.95** | **97.78%** |
| *Retain 160 Tokens in Average (↓ 94.4%), ~1.156 TFLOPs* | | | | | | | | | | | | | |
| FastV (Chen et al., 2024a) | *ECCV'24* | 48.19 | 47.96 | 39.77 | 1083.20 | 68.81 | 64.07 | 61.42 | 48.92 | 48.04 | 22.04 | 50.34 | 76.93% |
| TRIM (Song et al., 2025) | *COLING'25* | 57.36 | 61.53 | 47.79 | 1279.44 | 84.58 | 65.51 | 71.04 | 45.94 | 52.92 | 29.82 | 58.05 | 88.71% |
| VisionZip (Yang et al., 2025b) | *CVPR'25* | 56.28 | 59.72 | 51.33 | 1342.84 | 84.06 | 65.89 | 73.49 | 54.32 | 53.68 | 34.59 | 60.05 | 91.78% |
| DivPrune (Alvar et al., 2025) | *CVPR'25* | 55.62 | 62.97 | 53.51 | 1359.82 | 81.05 | 66.28 | 74.48 | 54.18 | 54.87 | 32.38 | 60.33 | 92.21% |
| SparseVLM (Zhang et al., 2024c) | *ICML'25* | 53.27 | 62.93 | 53.45 | 1362.86 | 84.83 | 66.08 | 71.83 | 53.85 | 54.61 | 34.81 | 60.38 | 92.28% |
| **Script (Ours)** | *Proposed* | **60.73** | **64.20** | **53.67** | **1423.89** | **87.19** | **67.44** | **76.71** | **55.28** | **55.19** | **36.38** | **62.80** | **95.98%** |

**Greedy Token Selection via Kernel Decomposition** To select a compact yet informative subset of $k$ visual tokens, we perform approximate MAP (maximum a posteriori) inference under a $k$-DPP defined by the query-conditioned kernel $\tilde{L}$. This procedure allows us to find tokens that are not only individually relevant to the query but also collectively diverse in semantic content, reducing redundancy while ensuring broad query coverage. The objective is to select a subset $S$ of $k$ tokens such that the corresponding submatrix $\tilde{L}_S$ maximizes $\det(\tilde{L}_S)$, $\max_{|S|=k} \det\left(\tilde{L}_S\right)$. The determinant reflects both semantic relevance and diversity. A higher value indicates greater diversity in the selected subset.

To efficiently approximate the MAP solution[2] for DPP, we adopt the Cholesky-based greedy algorithm (Chen et al., 2018). The algorithm starts with an empty set and incrementally adds tokens. At each step, it selects the token that yields the highest gain, where the gain is defined as the increase in the determinant of the kernel submatrix after adding the token. The algorithm is initialized as follows: $v_i^2 = \tilde{L}_{ii}$, $\mathbf{u}_i = \mathbf{0} \in \mathbb{R}^k$ for all $i \in \{1, \ldots, n\}$, where $v_i^2$ represents the importance of token $i$ for selection, and $\mathbf{u}_i$ is a vector that stores the projection of token $i$ onto the subspace spanned by the previously selected tokens. At each iteration, the token with the largest $v^2$ is selected: $j = \arg\max_{i \in Z \setminus S} v_i^2$. After the selection, for each $i \notin (S \cup \{j\})$, we compute:

$$e_i = \frac{\tilde{L}_{ji} - \langle \mathbf{u}_j, \mathbf{u}_i \rangle}{\sqrt{v_j^2 + \epsilon}}, \quad u_i \leftarrow u_i + e_i \mathbf{e}_j, \quad v_i^2 \leftarrow v_i^2 - e_i^2, \tag{11}$$

where $e_i$ represents the projection of token $i$ onto the direction defined by token $j$, $\mathbf{e}_j$ is the $j$-th standard basis vector in $\mathbb{R}^n$, and $\epsilon$ is a small positive constant (e.g., $10^{-6}$) introduced for numerical stability. The update $v_i^2 \leftarrow v_i^2 - e_i^2$ encourages selecting tokens with small projections $e_i$. A smaller $e_i$ indicates that the token is less similar, that is, closer to being orthogonal to the already selected tokens, thereby enhancing the diversity of the subset.

## 4.3 Final Token Selection

GSP and QCSP each address complementary aspects of token selection. GSP explicitly targets visual redundancy, while QCSP captures query relevance. However, neither is sufficient on its own for robust selection. We therefore take their intersection to ensure that the final subset is both visually compact and semantically aligned with the query. If the intersection contains fewer tokens than the required number, we supplement it with additional tokens from QCSP by iteratively increasing $k$ to retrieve more tokens.

---

[2]MAP for DPP is an NP-hard problem.

Table 4: **Performance comparison of different pruning methods on Video-LLaVA-7B with 64 frames per video.** The best results in each setting are **bolded**, and the second-best are underlined.

| Method | Venue | MLVU | MVBench | LongVideoBench | | | Video-MME | | | | Acc. | Relative |
|---|---|---|---|---|---|---|---|---|---|---|---|---|
| Metric | | m-avg | test | val | perception | relation | w/o sub | short | medium | long | | |
| *Upper Bound, All 64 × 169 Tokens (100%)* | | | | | | | | | | | | |
| LLaVA-Video-7B | - | 67.75 | 58.15 | 59.07 | 65.03 | 53.64 | 63.66 | 76.46 | 61.21 | 53.19 | 62.19 | 100% |
| *Retain 64 × 64 Tokens (↓ 62.1%)* | | | | | | | | | | | | |
| FastV Chen et al. (2024a) | *ECCV'24* | 63.79 | 55.81 | 56.21 | 60.60 | 52.15 | 61.89 | 73.46 | 59.45 | **52.71** | 59.56 | 95.77% |
| DivPrune Alvar et al. (2025) | *CVPR'25* | 64.01 | 54.82 | 57.97 | **64.32** | **53.75** | 61.31 | 72.49 | 59.43 | 51.02 | 59.90 | 96.32% |
| SparseVLM Zhang et al. (2024c) | *ICML'25* | 65.33 | 56.68 | 56.30 | 61.20 | 51.57 | 60.94 | 73.20 | 58.78 | 51.12 | 59.45 | 95.59% |
| **Script (Ours)** | *Purpose* | **66.13** | **57.74** | **58.27** | 64.25 | 53.37 | **62.36** | **74.67** | **60.31** | 52.59 | **61.08** | **98.22%** |
| *Retain 64 × 32 Tokens (↓ 81.1%)* | | | | | | | | | | | | |
| FastV Chen et al. (2024a) | *ECCV'24* | 58.75 | 52.27 | 52.64 | 56.88 | 48.25 | 56.00 | 63.38 | 55.79 | 48.34 | 54.70 | 87.95% |
| DivPrune Alvar et al. (2025) | *CVPR'25* | 61.44 | 53.37 | **56.64** | **62.41** | 51.34 | 59.73 | 69.69 | 57.82 | 49.88 | 58.04 | 93.32% |
| SparseVLM Zhang et al. (2024c) | *ICML'25* | 60.37 | 54.13 | 53.47 | 58.51 | 49.79 | 59.09 | 69.18 | 56.49 | 50.38 | 56.82 | 91.37% |
| **Script (Ours)** | *Purpose* | **62.77** | **55.37** | 56.35 | 61.40 | **52.97** | **60.35** | **72.11** | **58.46** | **51.20** | **58.99** | **95.00%** |
| *Retain 64 × 16 Tokens (↓ 90.5%)* | | | | | | | | | | | | |
| FastV Chen et al. (2024a) | *ECCV'24* | 52.58 | 46.57 | 46.61 | 48.68 | 44.79 | 49.88 | 54.92 | 50.30 | 45.28 | 48.85 | 78.54% |
| DivPrune Alvar et al. (2025) | *CVPR'25* | 58.60 | 51.13 | 52.27 | **57.56** | 47.29 | 56.37 | **67.67** | 53.34 | 48.11 | 54.72 | 88.98% |
| SparseVLM Zhang et al. (2024c) | *ICML'25* | 51.97 | 48.87 | 47.42 | 53.02 | 42.89 | 49.58 | 53.81 | 49.45 | 46.22 | 49.25 | 79.19% |
| **Script (Ours)** | *Purpose* | **58.77** | **53.45** | **52.91** | 57.13 | **48.66** | **57.20** | 65.97 | **56.08** | **49.46** | **55.52** | **89.30%** |

# 5 Experiments

## 5.1 Experimental Settings

We conduct comprehensive experiments to evaluate Script's performance across diverse MLLMs and benchmarks[3]. In all tables, the "Relative" column reports performance relative to the unpruned model.

**Models and Baselines.** We select four representative MLLMs: LLaVA 1.5 7B (Liu et al., 2023), LLaVA NeXT 7B (Liu et al., 2024a), Video LLaVA 7B (Lin et al., 2024), and Intern VL3 (Zhu et al., 2025). These models cover both image and video modalities with diverse architectures, providing a comprehensive testbed for evaluating Script's generalizability. In addition, we compare with five state-of-the-art pruning baselines, including FastV (Chen et al., 2024a), TRIM (Song et al., 2025), SparseVLM (Zhang et al., 2024c), DivPrune (Alvar et al., 2025), and VisionZip (Yang et al., 2025b). Notably, most of these baselines are either query-agnostic or tightly bound to specific model architectures. Some of these architectures are included in our evaluation, enabling fair and direct comparisons.

**Benchmarks.** We evaluate Script across fourteen widely adopted benchmarks spanning both image and video understanding tasks. **Image understanding tasks:** GQA (Hudson & Manning, 2019), ScienceQA (Lu et al., 2022), VQAv2 (Goyal et al., 2017), TextVQA (Singh et al., 2019), VizWiz (Gurari et al., 2018), MMVet (Yu et al., 2023), MMBench (Liu et al., 2024c), MMBench-CN (Liu et al., 2024c), MME (Fu et al., 2023), and POPE (Li et al., 2023b). **Video reasoning tasks:** LongVideoBench (Wu et al., 2024), VideoMME (Fu et al., 2025), MLVU (Zhou et al., 2025), and MVBench (Li et al., 2023a).

## 5.2 Results and Discussions

**LLaVA-1.5-7B.** Table 2 compares visual token pruning methods on LLaVA-1.5-7B across different token budgets. Our proposed method, Script, consistently outperforms baselines such as FastV, DivPrune, and SparseVLM, demonstrating strong adaptability and robustness across pruning levels and diverse tasks. At the highest token budget (192 tokens), Script achieves perfect accuracy (100.00%), outperforming the next-best method, DivPrune, by over 4 percentage points (95.94%). When reducing tokens to an intermediate level (64 tokens), Script retains exceptional accuracy at 96.86%, indicating a minimal performance drop despite significant token reduction. This outperforms SparseVLM by around 6 percentage points, underscoring Script's effective token selection strategy. Even under the most aggressive pruning (16 tokens, corresponding

---

[3]See the appendix for additional experiments, model settings, limitations, and broader impact.

to 97.2% pruning), Script still retains high accuracy (89.51%), significantly outperforming all baselines, highlighting its strong ability to preserve essential semantic and visual information, making it suitable for resource-constrained deployment.

**LLaVA-NeXT-7B.** Table 3 presents a detailed comparison of visual token pruning methods applied to LLaVA-NeXT-7B across different token budgets. Our method, Script, consistently delivers the best performance across pruning levels, outperforming all baseline methods. Under mild pruning (640 tokens, 77.8% reduction), Script achieves the highest average accuracy (65.1%) and the best relative performance (99.88%), indicating minimal information loss despite significant pruning. Under moderate pruning conditions (320 tokens, 88.9% reduction), Script maintains robust performance, achieving 63.97% accuracy and 97.09% relative performance, outperforming the next-best baseline by 2.12% points. Even at the most aggressive pruning level (160 tokens, 94.4% reduction), Script still maintains superior results, reaching 62.80% accuracy and 95.98% relative performance. These results demonstrate Script's strong ability to retain essential visual and semantic cues under tight computational budgets.

**Video-LLaVA-7B.** Table 4 reports the performance of SOTA pruning methods on Video-LLaVA-7B across multiple token budgets. Script consistently outperforms all baselines across pruning levels, demonstrating robust performance under growing compression ratios. Under mild pruning ($64{\times}64$ tokens, 62.1% reduction), Script achieves 98.22% average accuracy, demonstrating effective preservation of informative video content essential for accurate multimodal reasoning. Furthermore, under moderate pruning ($64{\times}32$ tokens, 81.1% reduction), Script maintains 95.00% accuracy, outperforming FastV by roughly 7 percentage points, which demonstrates its effectiveness even under tighter token constraints. Even under aggressive pruning ($64{\times}16$ tokens, 90.5% reduction), Script retains strong performance with 89.30% average accuracy, significantly surpassing other baselines at similar compression levels. These results highlight Script's ability to preserve spatio-temporal semantics, supporting its applicability in real-world video-based multimodal tasks.

## 5.3 Ablation study

To analyze the individual impact of each module, we conduct an ablation study to evaluate the contributions of the GSP, QCRS, and DPP modules. All experiments are conducted under stringent constraints, retaining only 16 tokens on average (97.3% pruning), with a compute cost of 0.103 TFLOPs. As shown in Table 5, naively pruning tokens causes a sharp drop in relative accuracy (average: 63.73%), underscoring the need for informed token selection strategies. Incorporating GSP alone raises relative accuracy to 66.58%,

Table 5: Ablation on LLava-1.5-7B evaluating the impact of GSP, QCRS, and DPP.

| GSP | QCRS | DPP | POPE | MME | GQA | SQA$^{IMG}$ | Acc. | Relative |
|---|---|---|---|---|---|---|---|---|
| *LLaVA-1.5-7B, Retain 576 Tokens (100%)* | | | | | | | | |
| | | | 86.96 | 1507.06 | 61.94 | 69.41 | 73.41 | 100% |
| *Retain 16 Tokens in Average ($\downarrow$ 97.3%), $\sim$0.103 TFLOPs* | | | | | | | | |
| ✘ | ✘ | ✘ | 49.92 | 695.70 | 38.69 | 63.76 | 46.78 | 63.73% |
| ✔ | ✘ | ✘ | 52.85 | 758.23 | 40.86 | 63.86 | 48.87 | 66.58% |
| ✘ | ✔ | ✘ | 77.20 | 1225.27 | 45.82 | 68.67 | 63.23 | 86.14% |
| ✔ | ✔ | ✘ | 77.75 | 1111.92 | 51.11 | 68.57 | 63.25 | 86.17% |
| ✘ | ✔ | ✔ | 83.38 | 1252.28 | 53.28 | 68.51 | 66.95 | 91.19% |
| ✔ | ✔ | ✔ | **84.44** | **1258.55** | **54.42** | **69.12** | **67.73** | **92.19%** |

demonstrating the benefit of using a bipartite graph to model redundancy. Integrating QCSP (QCRS + DPP) alone further improves performance, increasing relative accuracy to 91.19%. Notably, the full configuration achieves 92.19% of the original accuracy while retaining only 2.7% of tokens. These results further reflect the distinct yet complementary contributions of each module.

## 5.4 Case study

To analyze how different pruning strategies handle query-relevant vision tokens, we compare four approaches: attention-based (via [CLS]), divergence-based, similarity-based, and DPP-based. Figure 5 visualizes the normalized scores from each method on a representative POPE benchmark. The scores reveal how each method prioritizes tokens based on distinct selection criteria.

As shown in Figure 5, the four strategies exhibit clearly different token selection behaviors. The attention-based method emphasizes globally salient regions, such as central objects, but often overlooks contextually relevant backgrounds like human bodies. In contrast, divergence-based and similarity-based methods struggle to prioritize tokens that align with the input query. The similarity-based method, in particular, ignores query information and produces relatively uniform token scores that primarily reflect visual diversity. While such diversity helps preserve overall image content, it fails to capture query-specific relevance. The DPP-based

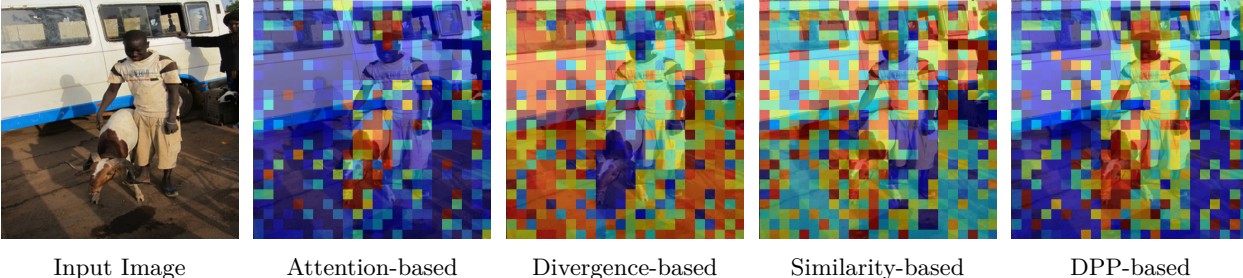

| Input Image | Attention-based | Divergence-based | Similarity-based | DPP-based |

Figure 5: **Visualizations of Pruning Preferences.** We compute four different pruning scores for a sample from the POPE benchmark using LLaVA-1.5-7B, with the query: *"Is there a person in the image?"* **Red** indicates high preference, while **blue** indicates low.

method assigns the highest relevance to human faces and moderately high scores to body-related tokens, demonstrating strong alignment with query semantics. However, similar to attention-based approaches, it lacks diversity in the selected tokens. These observations highlight the importance of combining GSP and QCSP for more balanced and effective token pruning.

## 5.5 Efficiency Analysis

We comprehensively evaluate recent token pruning methods on the POPE benchmark, focusing on computation cost, inference latency, memory usage, and accuracy. As shown in Table 6, the uncompressed LLaVA-NeXT-7B requires 41.7T FLOPs, with 246ms prefill and 29ms decode latency per token. It consumes 1440MB KV cache and 16.7GB peak GPU memory, achieving an F1 score of 86.8. In con-

Table 6: **Efficiency comparisons on the POPE benchmark**. We report the theoretical FLOPs, actual runtime, KV cache compression rate (%), and the achieved accuracy.

| Method | # Token | FLOPs (T) | Prefill Time (ms/token) | Decode Time (ms/token) | KV Cache (MB) | GPU Memory (GB) | F1 Score (F1) |
|---|---|---|---|---|---|---|---|
| LLaVA-NeXT-7B | 2880 | 41.7 | 246 | 29 | 1440.0 | 16.7 | 86.8 |
| FastV(ECCV24) | 320 | 4.4 (×9.5) | 54 (×4.6) | 23 (×1.2) | 160.3 | 15.6 | 49.5 |
| PDrop(CVPR25) | 320 | 4.5 (×9.3) | 55 (×4.5) | 24 (×1.2) | 160.2 | 15.6 | 60.8 |
| SparseVLM(ICML25) | 320 | 4.5 (×9.3) | 71 (×3.5) | 25 (×1.1) | 161.2 | 18.6 | 76.9 |
| VisionZip(CVPR25) | 320 | 4.2 (×9.9) | 38 (×6.6) | **22** (×1.3) | **160.0** | 14.8 | 82.3 |
| DivPrune(CVPR25) | 320 | 4.2 (×9.9) | 38 (×6.6) | **22** (×1.3) | **160.0** | 13.9 | 84.7 |
| **Script**(Ours) | 320 | **4.1** (×10) | **35** (×6.8) | **22** (×1.3) | **160.0** | **13.5** | **86.7** |
| w/o DPP | 320 | **4.1** (×10) | 38 (×6.6) | **22** (×1.3) | **160.0** | 14.0 | 86.5 |

trast, Script reduces FLOPs to 4.1T (10× reduction), prefill latency to 35ms (6.8× faster), and decode latency to 22ms. It also lowers KV cache to 160MB and memory to 13.5GB, while maintaining an F1 score of 86.7. Without DPP, Script maintains similar efficiency and achieves a slightly lower F1 score of 86.5. Compared to other baselines, Script achieves the lowest prefill/decode latency, the smallest KV cache, and the best overall F1 score. For instance, FastV and PDrop suffer major accuracy loss (F1: 49.5 and 60.8), while VisionZip and DivPrune perform better (F1: 82.3 and 84.7) but at higher computational and memory cost. Overall, Script achieves the best balance between efficiency and accuracy, outperforming all baselines across nearly all metrics.

## 6 Conclusion

In this paper, we introduce **Script** which reduces inference cost by selecting a compact yet semantically meaningful subset of visual tokens, conditioned on the user query. Script combines visual redundancy and query relevance perspectives through four stages: (1) graph-structured filtering to remove visually redundant tokens, (2) query-conditioned relevance scoring, (3) diversity-promoting subset selection based on relevance scores using DPP, and (4) intersection-based fusion to retain tokens that are both visually compact and semantically aligned with the query. Its architecture-agnostic, plug-and-play design enables seamless integration into existing MLLMs without requiring retraining or architectural changes. Extensive experiments on fourteen vision-language benchmarks show that Script consistently reduces computational overhead and latency while preserving, or even improving, task performance under aggressive pruning settings. Future work includes extending Script to additional modalities such as audio and video in spatial-temporal contexts, and exploring adaptive pruning schedules that dynamically adjust to input complexity.

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

# Script: Graph-Structured and Query-Conditioned Semantic Token Pruning for Multimodal Large Language Models Appendix

## A   Notation Overview

To facilitate clarity and improve readability, we provide a detailed glossary of all mathematical symbols used throughout this paper, as shown in Table A.1. The table serves as a centralized reference point, systematically organizing every notation involved in the core components of the Script framework, including equations, algorithmic modules, and architectural definitions. The glossary covers all key components of the proposed framework, including visual encoding, query representation, relevance scoring, similarity computation, DPP-based token selection, structural pruning, and auxiliary constants. This table is intended to improve the clarity of mathematical derivations, reduce cross-referencing overhead, and support accurate reproduction and further development of the method.

Table A.1: Detailed symbol glossary used throughout the *Script* framework.

| Symbol | Type / Dim | Definition |
|---|---|---|
| $\boldsymbol{X_v}$ | Image / video input | Raw visual input passed into the vision encoder $f_v$. |
| $f_v$ | Function | Vision encoder extracting patch-level features from $\boldsymbol{X_v}$. |
| $g$ | Function | Multimodal projector mapping visual features to token embeddings. |
| $n, d, \ell, m$ | Integers | #tokens (input, dim, query length, retained) respectively. |
| $\boldsymbol{H_v}$ | Matrix | Sequence of $n$ visual tokens, each $d$-dimensional, before pruning. |
| $\boldsymbol{H_q}$ | Sequence of vectors | Tokenized user query of length $\ell$. |
| $\tilde{\boldsymbol{H}}_v$ | Matrix ($m \times d$) | Visual token subset after pruning ($m < n$). |
| $f_\varphi$ | Function (LLM) | Frozen language model consuming $[\tilde{\boldsymbol{H}}_v; \boldsymbol{H_q}]$. |
| $L(\cdot, \cdot)$ | Scalar loss | Task loss comparing model outputs before and after pruning. |
| $\mathbf{h}_i$ | Vector ($\in \mathbb{R}^d$) | $i$-th visual token embedding in $\boldsymbol{H_v}$. |
| $\mathbf{h}_\mu^{(q)}$ | Vector ($\in \mathbb{R}^d$) | Mean pooled embedding of the query tokens. |
| $q_i$ | Scalar | Query-conditioned relevance score of token $i$. |
| $\boldsymbol{Q} = \mathrm{diag}(q_1, \ldots, q_n)$ | Diagonal matrix | Relevance weighting matrix for visual tokens. |
| $S_{ij}$ | Scalar | Cosine similarity between tokens $i$ and $j$. |
| $\boldsymbol{S}$ | Matrix ($n \times n$) | Symmetric similarity matrix among visual tokens. |
| $\boldsymbol{L} = \boldsymbol{Q}^{1/2} \boldsymbol{S} \boldsymbol{Q}^{1/2}$ | Matrix ($n \times n$) | DPP kernel combining relevance and diversity. |
| $\mathcal{I}, \; |\mathcal{I}| = m$ | Index set | Token indices selected for retention. |
| $\mathcal{I}^*$ | Index set | Optimal index subset maximizing log-det of DPP kernel. |
| $\tilde{\boldsymbol{L}}$ | Matrix | Query-conditioned DPP kernel with temperature scaling. |
| $\mathbf{r}, \tilde{\mathbf{r}}$ | Vectors ($\in [0, 1]^n$) | Raw and temperature-scaled relevance scores. |
| $V_{\mathrm{src}}, V_{\mathrm{dst}}$ | Token subsets | Bipartite partitions used in graph-based redundancy pruning. |
| $G = (V_{\mathrm{src}}, V_{\mathrm{dst}}, \boldsymbol{S})$ | Graph | Token similarity graph for structural filtering. |
| $d_i^2$ | Scalar | Residual variance of token $i$ under low-rank projection. |
| $u_i$ | Vector ($\in \mathbb{R}^T$) | Coefficient vector for token $i$ in basis space. |
| $e_i$ | Scalar | Normalizer in Cholesky-style update. |
| $\boldsymbol{C} \in \mathbb{R}^n \times T$ | Matrix | Basis matrix for low-rank approximation of DPP kernel. |
| $T$ | Integer | Target rank or DPP subset size. |
| $\tau$ | Threshold (scalar) | Redundancy threshold for similarity pruning. |
| $\epsilon$ | Constant ($10^{-6}$) | Numerical stability term in projection update. |
| $\odot$ | Operator | Element-wise (Hadamard) product. |
| $\langle \cdot, \cdot \rangle$ | Operator | Inner product used for cosine similarity. |
| $\|\cdot\|$ | Operator | Euclidean norm used in normalization. |

# B  Diversity in Determinantal Point Processes: Geometric Intuition and Theoretical Analysis

## B.1  Notation.

To facilitate clear and consistent exposition, we first define the key symbols and assumptions used throughout the paper and illustrate each with a simple example or explanation.

- $\|\cdot\|_2$ **(Euclidean norm):** The standard $\ell_2$ norm for vectors in $\mathbb{R}^d$.
  *Example:* $\|(3,4)\|_2 = \sqrt{3^2 + 4^2} = 5$.

- $\lambda_i(A)$ **($i$-th largest eigenvalue of $A$):** For a symmetric matrix $A \in \mathbb{R}^{k \times k}$, we order its eigenvalues as $\lambda_1(A) \geq \lambda_2(A) \geq \cdots \geq \lambda_k(A)$. Since all our kernels will be positive semi-definite (PSD), all $\lambda_i \geq 0$.

- $\mathbf{1}_k$ **(all-ones vector):** A $k$-dimensional column vector with all entries equal to one, i.e., $(1, \ldots, 1)^\top \in \mathbb{R}^k$.

Let $\mathcal{T} = v_1, \ldots, v_n \subset \mathbb{R}^d$ denote a collection of feature vectors. We assume that each $v_i$ has unit norm: $\|v_i\|_2 = 1$ for all $i$. This means that each vector lies on the unit hypersphere in $\mathbb{R}^d$.

*Example:* The vector $v_i = (1, 0, \ldots, 0)^\top$ has $\|v_i\|_2 = 1$, and thus is a valid member of $\mathcal{T}$.

We adopt this unit-norm convention to ensure that inner products $v_i^\top v_j$ correspond exactly to cosine similarities, simplifying the interpretation of the similarity kernel introduced next. Meanwhile, this normalization remains fixed throughout the paper.

**Similarity Kernel.** Given a collection of unit-norm vectors $\mathcal{T} = [v_1, \ldots, v_n] \subset \mathbb{R}^d$ as introduced above, we define a similarity kernel matrix $L \in \mathbb{R}^{n \times n}$ whose $(i, j)$-entry captures the cosine similarity between $v_i$ and $v_j$:

$$L_{ij} = v_i^\top v_j, \qquad \text{for all } 1 \leq i, j \leq n. \tag{B.1}$$

Since all $v_i$ lie on the unit hypersphere, we have $v_i^\top v_j \in [-1, 1]$. In particular, $L_{ii} = v_i^\top v_i = 1$, and the more similar two vectors are, the closer $L_{ij}$ is to 1.

We can express the full kernel compactly using matrix notation. Let $V = [v_1 \ v_2 \ \cdots \ v_n] \in \mathbb{R}^{d \times n}$ be the matrix whose columns are the feature vectors.

Then:

$$L = V^\top V. \tag{B.2}$$

This representation immediately implies that $L$ is symmetric and positive semi-definite (PSD), because for any $x \in \mathbb{R}^n$, we have:

$$x^\top L x = x^\top V^\top V x = \|V x\|_2^2 \geq 0. \tag{B.3}$$

Therefore, $L$ satisfies the standard spectral properties of Gram matrices: all eigenvalues of $L$ are real and non-negative, and $L$ admits an orthonormal eigendecomposition. These properties are critical for the DPP model discussed next.

## B.2  Determinant as a Diversity Objective

We now connect the determinant of a kernel submatrix with geometric diversity. Specifically, we show that for any subset of $k$ unit vectors, the determinant of their Gram matrix equals the square of the volume of the parallelotope they span.

**Proposition 1** (Determinant–Volume Equivalence). *Let $S = [i_1, \ldots, i_k] \subseteq [n]$ be an index subset of size $k$. Define $V_S = [v_{i_1}, \ldots, v_{i_k}] \in \mathbb{R}^{d \times k}$ as the submatrix of feature vectors indexed by $S$, and $L_S = V_S^\top V_S \in \mathbb{R}^{k \times k}$ as their Gram matrix. Then,*

$$\det L_S = \det(V_S^\top V_S) = (\mathrm{Vol}_k(V_S))^2, \tag{B.4}$$

where $\mathrm{Vol}_k(V_S)$ *is the $k$-dimensional volume of the parallelotope spanned by the column vectors of $V_S$.*

*Proof.* Let $V_S = QR$ be the QR decomposition of $V_S$, where $Q \in \mathbb{R}^{d \times k}$ has orthonormal columns and $R \in \mathbb{R}^{k \times k}$ is upper triangular. Then,

$$L_S = V_S^\top V_S = R^\top Q^\top Q R = R^\top R, \tag{B.5}$$

using the fact that $Q^\top Q = I_k$. Taking determinants on both sides gives:

$$\det L_S = \det(R^\top R) = (\det R)^2. \tag{B.6}$$

Geometrically, the $k$-dimensional volume satisfies

$$\mathrm{Vol}_k(V_S) = \sqrt{\det(V_S^\top V_S)} = |\det R|, \tag{B.7}$$

hence $\det L_S = (\mathrm{Vol}_k(V_S))^2$. $\qquad\square$

This equivalence forms the foundation for interpreting $\det L_S$ as a natural diversity objective: it assigns higher values to sets of vectors that are more 'spread out' or 'orthogonal' in space.

### B.3 Diversity Upper Bound via Hadamard Inequality

The determinant of any Gram matrix $L_S$ formed from unit-norm vectors is upper bounded by 1. This follows from the classical Hadamard inequality, which gives a constraint on the determinant of a positive semi-definite matrix in terms of its diagonal entries.

**Corollary 1** (Hadamard Bound). *Let $L_S$ be the Gram matrix defined above, formed from $k$ unit vectors. Then:*

$$\det L_S \leq 1, \qquad \det L_S = 1 \iff v_{i_p}^\top v_{i_q} = 0 \text{ for all } p \neq q \text{ and } k \leq d. \tag{B.8}$$

*Proof.* Hadamard's inequality states that for any positive semi-definite matrix $A \in \mathbb{R}^{k \times k}$ with diagonal entries $a_{ii}$,

$$\det A \leq a_{11} a_{22} \cdots a_{kk},$$

with equality if and only if the columns of $A$ are orthogonal.

In our case, $L_S = V_S^\top V_S$ is PSD and satisfies $\mathrm{diag}(L_S) = (1, 1, \ldots, 1)$ since each $\|v_i\|_2 = 1$.

Therefore,

$$\det L_S \leq 1, \tag{B.9}$$

with equality if and only if the vectors $v_{i_1}, \ldots, v_{i_k}$ are mutually orthogonal; this requires $k \leq d$. If $k > d$, then $\mathrm{rank}(L_S) \leq d < k$ and $\det L_S = 0 < 1$. $\qquad\square$

The determinant $\det L_S$ quantifies how linearly independent or "spread out" the vectors in $S$ are. The maximum value 1 occurs when the vectors are exactly orthogonal. Therefore, maximizing $\det L_S$ promotes diversity by encouraging the selection of vectors that are as close to orthogonal as possible.

### B.4 Determinantal Point Processes and Diversity Maximization

We now formally introduce Determinantal Point Processes (DPPs), which are probability distributions over subsets that inherently promote diversity. In the fixed-size or $k$-DPP setting, each subset $S$ of size $k$ is assigned a probability proportional to the determinant of its associated submatrix $L_S$:

$$\mathbb{P}_L(S) \propto \det L_S, \qquad \text{for } |S| = k. \tag{B.10}$$

Specifically, we formulate this as:

$$\mathbb{P}_L(S) = \frac{\det L_S}{\sum_{\substack{T \subseteq [n] \\ |T|=k}} \det L_T}. \tag{B.11}$$

where the denominator sums over all $k$-subsets of $[n]$, ensuring a valid probability distribution.

This probabilistic model encourages the selection of sets $S$ whose vectors are geometrically diverse, since higher determinant values correspond to higher volume (as shown previously). In practice, one often seeks the *most probable* subset under this model, known as the maximum a posteriori (MAP) estimate:

$$S_{\mathrm{MAP}} = \arg\max_{|S|=k} \det L_S. \tag{B.12}$$

This optimization problem aims to find the subset of $k$ vectors that span the largest volume parallelotope in $\mathbb{R}^d$, or equivalently, the subset exhibiting maximal geometric diversity. The DPP framework thus provides both a probabilistic model and a concrete objective—$\det L_S$—for achieving diverse subset selection.

## B.5 Redundancy Metrics and Determinant Bounds

While the determinant $\det L_S$ captures diversity, we can also assess subset quality via *redundancy* metrics. These measure how similar the vectors in $S$ are to each other, either in the worst-case or on average. Two commonly used metrics are:

$$\rho_{\max}(S) = \max_{i \neq j \in S} L_{ij}, \qquad \rho_{\mathrm{avg}}(S) = \frac{2}{k(k-1)} \sum_{i<j \in S} L_{ij}. \tag{B.13}$$

Here, $L_{ij} = v_i^\top v_j$ is the cosine similarity between unit vectors $v_i$ and $v_j$. The value $\rho_{\max}(S)$ measures the most redundant pair (i.e., most similar), while $\rho_{\mathrm{avg}}(S)$ captures the overall similarity level.

We now relate these redundancy measures to $\det L_S$ using spectral bounds. The first bound is derived using Gershgorin's circle theorem.

**Lemma 1** (Gershgorin-Based Lower Bound). *For any subset $S$ of size $k$, define $\rho_\infty(S) := \max_{i \neq j \in S} |L_{ij}|$. Then:*

$$\lambda_{\min}(L_S) \geq 1 - (k-1)\,\rho_\infty(S), \qquad \det L_S \geq [1 - (k-1)\,\rho_\infty(S)]_+^k. \tag{B.14}$$

*Proof.* For each row $i$ of $L_S$, the Gershgorin radius is $r_i = \sum_{j \neq i} |L_{ij}| \leq (k-1)\rho_\infty(S)$. By Gershgorin's circle theorem, every eigenvalue of $L_S$ lies in at least one disk centered at 1 with radius $r_i$, hence

$$\lambda_{\min}(L_S) \geq 1 - \max_i r_i \geq 1 - (k-1)\rho_\infty(S).$$

Since $L_S$ is symmetric PSD, its determinant is the product of eigenvalues, so

$$\det L_S = \prod_{i=1}^{k} \lambda_i(L_S) \geq (\lambda_{\min}(L_S))^k \geq [1 - (k-1)\rho_\infty(S)]_+^k.$$

$\square$

Next, we upper bound $\det L_S$ via the arithmetic mean–geometric mean inequality.

**Lemma 2** (AM–GM Upper Bound). *Let $\lambda_1, \ldots, \lambda_k$ be the eigenvalues of $L_S$. Then:*

$$\det L_S \leq \left( \frac{\mathrm{tr}(L_S)}{k} \right)^k = 1. \tag{B.15}$$

**Discussion.** This bound follows directly from the arithmetic mean–geometric mean inequality: the product of nonnegative eigenvalues is at most the $k$-th power of their average. Since $L_S$ is a Gram matrix of unit-norm vectors, all diagonal entries are 1, so $\mathrm{tr}(L_S) = k$. Hence, the right-hand side reduces to 1, independent of the actual off-diagonal similarities.

Therefore, the AM–GM bound is extremely loose: it only states the trivial fact that $\det L_S \leq 1$, without reflecting the effect of redundancy or $\rho_{\mathrm{avg}}(S)$. This motivates the refined spectral bound derived next, which captures how $\det L_S$ decays with increasing average similarity.

**Motivation for a Refined Bound.** The AM–GM bound above provides a worst-case envelope that does not incorporate any spectral structure of $L_S$ beyond its average similarity. However, when all off-diagonal entries of $L_S$ are equal to $\rho_{\mathrm{avg}}$, the matrix exhibits an extremal spectral configuration that *maximizes* the determinant under a fixed average similarity. By explicitly analyzing this case, we obtain a refined *upper envelope* that better captures how the maximum achievable $\det L_S$ decays with increasing redundancy.

**Lemma 3** (Refined Upper Bound via Spectral Construction). *Let $L_S$ be the Gram matrix of $k$ unit vectors with average off-diagonal similarity $\rho_{\mathrm{avg}}(S) \in [-\frac{1}{k-1}, 1)$. Then,*

$$\det L_S \leq (1 + (k-1)\rho_{\mathrm{avg}}(S)) \cdot (1 - \rho_{\mathrm{avg}}(S))^{k-1}. \tag{B.16}$$

*Proof.* Consider the class of symmetric PSD matrices $L_S$ where all off-diagonal entries are equal to $\rho_{\mathrm{avg}}$, and all diagonal entries are 1:

$$(L_S)_{ij} = \begin{cases} 1 & \text{if } i = j \\ \rho_{\mathrm{avg}} & \text{if } i \neq j, \end{cases}$$

which is feasible iff $\rho_{\mathrm{avg}} \in [-\frac{1}{k-1}, 1)$. This matrix has a known spectral decomposition: one eigenvalue equals $1 + (k-1)\rho_{\mathrm{avg}}$, and the remaining $(k-1)$ eigenvalues are $1 - \rho_{\mathrm{avg}}$.

Hence, the determinant is:

$$\det L_S = (1 + (k-1)\rho_{\mathrm{avg}}) \cdot (1 - \rho_{\mathrm{avg}})^{k-1}.$$

Among all PSD matrices with unit diagonal and a fixed average off-diagonal similarity, the functional $\log \det(\cdot)$ is concave and permutation-invariant; by symmetry, an optimizer (for maximizing det under this constraint) must be permutation-invariant, i.e., the equicorrelation matrix above. Thus, the displayed value yields an upper bound, establishing the claim. $\square$

This refined bound is an upper bound for all positive semi-definite matrices with unit diagonal and a given average off-diagonal similarity, and it is *tight*: the equicorrelation matrix attains equality. Among such uniformly redundant matrices, the bound is achieved exactly, making it a sharp envelope for assessing diversity degradation as average correlation increases.

Compared to Lemma 2, this refined bound is strictly tighter whenever $\rho_{\mathrm{avg}}(S) > 0$, as it directly models the spectral behavior of uniformly redundant configurations. The bound thus provides a sharper envelope for analyzing diversity degradation in highly correlated subsets.

### B.6 Global Optimality of the MAP Subset

**Proposition 2** (MAP subset need not minimize redundancy). *In general, a $k$-subset $S^* = \arg\max_{|S|=k} \det L_S$ need not minimize either $\rho_{\mathrm{max}}$ or $\rho_{\mathrm{avg}}$ among all size-$k$ subsets.*

*Proof.* Consider $k = 2$ with

$$L = \begin{pmatrix} 1 & c \\ c & 1 \end{pmatrix}, \quad c \in [-1, 1].$$

Then $\det L = 1 - c^2$ is maximized at $c = 0$, where $\rho_{\mathrm{max}} = 0$. For $c = -\frac{1}{2}$, we have $\rho_{\mathrm{max}} = -\frac{1}{2} < 0$ (strictly smaller), yet $\det L = 1 - \frac{1}{4} = \frac{3}{4} < 1$. Hence maximizing $\det L$ does not generally minimize $\rho_{\mathrm{max}}$ (nor $\rho_{\mathrm{avg}}$). $\square$

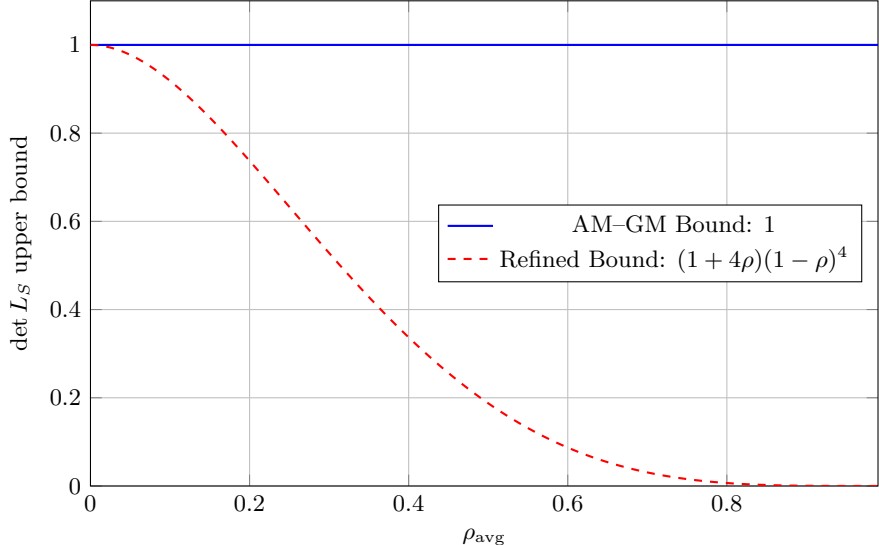

Figure B.1: Upper envelope on $\det L_S$ vs. average similarity $\rho_{\text{avg}}$ for $k = 5$. Unlike the trivial AM–GM bound (which is constantly 1), the refined envelope $(1 + 4\rho)(1 - \rho)^4$ decreases with redundancy, revealing how the maximum attainable determinant shrinks as $\rho_{\text{avg}}$ grows.

### B.7  Approximation via Greedy MAP Inference

Exact MAP inference for a $k$-DPP is NP-hard, as it requires evaluating $\binom{n}{k}$ determinants. However, the *regularized* objective

$$f(S) = \log \det(I + L_S)$$

is known to be monotone and submodular when $L$ is positive semi-definite, which allows for a natural greedy approximation algorithm. The function $\log \det(I + L_S)$ is monotone and submodular when $L$ is PSD, as it arises from the entropy of Gaussian variables or from spectral submodular theory.

Let $\widehat{S}$ denote the greedy solution obtained by iteratively selecting vectors that offer the greatest marginal increase in $\log \det(I + L_S)$. Then, by the classical Nemhauser–Wolsey theorem for submodular maximization:

$$\log \det(I + L_{\widehat{S}}) \geq (1 - 1/e) \log \det(I + L_{S^\star}), \tag{B.17}$$

where $S^\star$ is the exact optimal size-$k$ subset for $f(S) = \log \det(I + L_S)$.

Exponentiating both sides gives a multiplicative guarantee on the regularized determinant:

$$\det(I + L_{\widehat{S}}) \geq (\det(I + L_{S^\star}))^{1 - 1/e}. \tag{B.18}$$

Translating such guarantees into bounds on $\rho_{\text{max}}$ or $\rho_{\text{avg}}$ requires additional structural assumptions on $L$ (e.g., incoherence or equicorrelation); we therefore refrain from stating general redundancy guarantees here.

### B.8  Sufficient Condition for Equivalence

The counterexample in Proposition 2 shows that, in general, the MAP solution of a $k$-DPP need not minimize redundancy measures such as $\rho_{\text{max}}$ or $\rho_{\text{avg}}$. Nevertheless, under additional structural assumptions, maximizing geometric diversity and minimizing redundancy *do* become equivalent. We now state a sufficient condition under which this equivalence holds.

**Proposition 3** (Equivalence under Equicorrelation). *Suppose that for each candidate subset $S$ of size $k$, the associated Gram matrix $L_S$ has the form*

$$(L_S)_{ij} = \begin{cases} 1 & \text{if } i = j, \\ \rho & \text{if } i \neq j, \end{cases}$$

*for some $\rho \in [0, 1)$. Then:*

$$\arg \max_{|S|=k} \det L_S = \arg \min_{|S|=k} \rho_{\max}(S) = \arg \min_{|S|=k} \rho_{\text{avg}}(S).$$

*In other words, the MAP subset simultaneously maximizes geometric diversity and minimizes redundancy.*

*Proof.* For such an equicorrelation matrix, the eigenvalues are known in closed form:

$$\lambda_1 = 1 + (k-1)\rho, \qquad \lambda_2 = \cdots = \lambda_k = 1 - \rho.$$

Therefore,

$$\det L_S = \big(1 + (k-1)\rho\big)(1-\rho)^{k-1}.$$

Since $\rho_{\max}(S) = \rho_{\text{avg}}(S) = \rho$ in this setting, it suffices to analyze the monotonicity of $\det L_S$ with respect to $\rho$. Taking logarithms and differentiating,

$$\frac{d}{d\rho} \log \det L_S = \frac{k-1}{1+(k-1)\rho} - \frac{k-1}{1-\rho} < 0, \quad \text{for } \rho \in (0, 1).$$

Thus $\det L_S$ decreases strictly with $\rho$, and maximizing the determinant is equivalent to minimizing $\rho$. The claim follows. $\square$

This proposition establishes a clean sufficient condition under which DPP-based MAP inference exactly aligns with redundancy minimization: in equicorrelated settings, maximizing diversity volume and minimizing redundancy are equivalent optimization goals.

## B.9 Summary.

This section establishes a theoretical link between geometric diversity, measured through the determinant of a kernel (Gram) submatrix, and redundancy metrics such as $\rho_{\max}$ and $\rho_{\text{avg}}$. The analysis has revealed three central insights:

1. Maximizing $\det L_S$ systematically favors subsets of vectors that are nearly orthogonal to one another, thereby reducing both worst-case and average redundancy.

2. In general, the maximum a posteriori (MAP) solution of the $k$-DPP maximizes geometric diversity (via $\det L_S$), but it does not necessarily minimize $\rho_{\max}$ or $\rho_{\text{avg}}$; rather, large determinants enforce *envelope-type constraints* on admissible redundancy (Proposition 2).

3. However, under additional structure—most notably the *equicorrelation condition* formalized in Proposition 3—the MAP solution *does* coincide with redundancy minimization: maximizing $\det L_S$ is then equivalent to minimizing $\rho_{\max}$ and $\rho_{\text{avg}}$.

4. When exact MAP inference is computationally infeasible, a greedy algorithm on the monotone submodular surrogate $f(S) = \log \det(I + L_S)$ provides constant-factor guarantees on the regularized determinant, thereby preserving much of the diversity of the global optimum.

Taken together, these results justify the determinant objective as a principled and effective diversity metric, one that offers both rigorous theoretical foundations and practical algorithmic strategies for subset selection. This theoretical perspective provides a solid basis for applications where balancing diversity and redundancy is essential.

**Application to Script.** The diversity metric and DPP framework developed here directly ground the Script method proposed in the main text. Script is designed for a practical, query-conditioned multimodal setting, yet its core selection principle remains the same: maximize the determinant of a positive semidefinite kernel submatrix in order to favor diverse and complementary elements.

In Script, the kernel is defined as

$$L = \mathrm{diag}(r)\, S \,\mathrm{diag}(r), \tag{B.19}$$

where $S$ encodes token–token similarities (e.g., cosine similarity), and $r \in \mathbb{R}_{\geq 0}^n$ stores query-conditioned relevance scores. Because $S$ is symmetric positive semidefinite and $\mathrm{diag}(r)$ is diagonal with nonnegative entries, the resulting kernel $L$ remains positive semidefinite under the congruence transformation. This ensures that the spectral properties required for DPP inference are preserved, even when relevance weights vary across tokens.

Crucially, the token similarity structure in Script satisfies the equicorrelation-type condition analyzed in Proposition 3. Consequently, the MAP subset

$$S^* = \arg\max_{|S|=k} \det(L_S), \tag{B.20}$$

not only maximizes geometric diversity but also *simultaneously minimizes redundancy*. In other words, Script inherits the best of both worlds: the probabilistic diversity guarantee of DPPs and the redundancy minimization property under its structural assumptions.

This explains why Script can aggressively prune tokens while preserving both relevance and diversity. Empirical results presented in Section 5 confirm that maximizing $\det(L_S)$ with a query-conditioned kernel yields subsets that are simultaneously relevant, non-redundant, and diverse, thereby improving both summarization and retrieval performance.

# C   Details of experimental setup

## C.1   Model Architectures

**LLaVA-1.5 (Liu et al., 2023)** The LLaVA series represents a foundational line of open-source vision-language models (VLMs), recognized for their simple design, low training cost, and strong performance. The original LLaVA architecture integrates a pretrained CLIP (Radford et al., 2021) as the visual encoder and Vicuna (Chiang et al., 2023) as the language model, connected via a linear projection layer. This enables the LLM to accept image grid features as input. Through visual instruction tuning, LLaVA gains the ability to handle multimodal tasks. LLaVA-1.5 enhances this framework by replacing the linear connector with a multi-layer perceptron (MLP), increasing input image resolution, and utilizing a broader and more diverse set of instruction tuning data. These modifications lead to substantial performance improvements. The model processes images at a resolution of 336×336, resulting in 576 visual tokens per image.

**LLaVA-NeXT (Liu et al., 2024a)** LLaVA-NeXT (also referred to as LLaVA-1.6) builds upon LLaVA-1.5 by introducing a dynamic resolution mechanism aimed at improving visual perception. Instead of using a fixed resolution, the model selects an optimal aspect ratio based on the original image and increases its resolution by up to 4×. Importantly, the visual encoder remains unchanged. To handle the higher-resolution inputs, the image is divided into multiple sub-images of the original size. Each sub-image is encoded independently, and the resulting visual tokens are concatenated and passed to the language model. This approach enhances the model's performance in tasks such as visual reasoning, optical character recognition (OCR), and knowledge-intensive questions. For consistency and fair comparison, we fix the resolution to 672×672 (4× the original), generating 2,880 visual tokens per image.

**LLaVA-Video (Zhang et al., 2024d)** LLaVA-Video is a variant of the LLaVA family designed specifically for video understanding. It introduces the SlowFast frame sampling strategy to balance the number of frames and the density of visual tokens. The model utilizes SigLIP (Zhai et al., 2023) as the visual encoder and processes video frames at 384×384 resolution, encoding each frame into 729 visual tokens. To reduce computational load, a 2×2 average pooling operation is applied to the grid features, effectively reducing the number of visual tokens by a factor of 4. During evaluation, we uniformly sample 64 frames per video, resulting in a total of 10,816 visual tokens. This design allows LLaVA-Video to efficiently model both spatial and temporal aspects of visual input.

**InternVL3 (Zhu et al., 2025)** One of the most advanced open-source MLLMs at present. Building upon its predecessor, InternVL2.5, it retains the ViT-MLP-LLM architecture, integrating a Vision Transformer with a large language model through an MLP connector. InternVL3 features a native multimodal pre-training paradigm, jointly acquiring linguistic and multimodal capabilities in a single stage. It incorporates Variable Visual Position Encoding to handle extended multimodal contexts and employs advanced training techniques like supervised fine-tuning and mixed preference optimization. InternVL3 demonstrates superior performance across a wide range of multimodal tasks, including tool usage, GUI agents, industrial image analysis, and 3D vision perception.

## C.2   Evaluation Benchmarks

### C.2.1   General Image Benchmarks

**VQAv2 (Goyal et al., 2017)** An open-ended visual question answering benchmark that evaluates a model's ability to understand images, natural language, and commonsense knowledge. It contains 265,016 images from the COCO dataset (Lin et al., 2014) and abstract scenes, with each image paired with an average of 5.4 questions. Each question is annotated with 10 ground truth answers and 3 plausible alternatives. We use the test-dev split for evaluation.

**GQA (Hudson & Manning, 2019)** A large-scale and widely used VQA benchmark based on real-world images from the Visual Genome dataset (Krishna et al., 2017), specifically designed to test compositional reasoning and fine-grained visual understanding. It provides over 22 million balanced question-answer pairs, with each image accompanied by a detailed scene graph explicitly describing objects, attributes, and relationships. For our experiments, we evaluate on the standard test-dev balanced split.

**VizWiz (Gurari et al., 2018)** A real-world VQA benchmark created from images taken by blind users, paired with spoken questions and 10 crowd-annotated answers each. It introduces two main challenges: answering questions and detecting unanswerable ones, due to issues like poor image quality and ambiguous content. We use the test split for evaluation.

**ScienceQA (Lu et al., 2022)** A multimodal, multiple-choice QA benchmark covering diverse scientific domains. It includes 21,208 questions categorized across 26 topics, 127 categories, and 379 skills. Nearly half of the questions include image or text context, and the majority are supplemented with grounded lectures and detailed explanations. We evaluate using the test split for evaluation.

**POPE (Li et al., 2023b)** A benchmark focused on evaluating object hallucination in MLLMs. Using images from COCO, it formulates binary questions regarding the presence of specific objects in the scene. Precision, recall, and F1 score are used to quantify hallucination. We use the test split for evaluation.

**MME (Fu et al., 2023)** A broad benchmark assessing the perceptual and cognitive abilities of multimodal large language models comprises 14 subtasks across perception (e.g., counting, color, position, OCR) and cognition (e.g., commonsense reasoning, translation, code understanding). All binary instruction-answer pairs are manually constructed to avoid data leakage, ensuring rigorous evaluation.

**MMBench (Liu et al., 2024c)** A comprehensive benchmark designed to evaluate a wide range of multimodal capabilities. It features a large and diverse set of questions, surpassing prior benchmarks in scale and coverage. A novel CircularEval strategy, powered by ChatGPT, converts open-ended responses into structured formats for consistent scoring. A Chinese version, MMBench-CN, is also provided.

**MM-Vet (Yu et al., 2023)** A benchmark emphasizing the integration of diverse multimodal skills. It defines six core capabilities: recognition, OCR, knowledge, language generation, spatial reasoning, and mathematics, via 218 carefully designed challenging examples. Evaluation is conducted using ChatGPT to ensure consistency across varied real-world answer formats.

**HallusionBench (Guan et al., 2024)** An image-context reasoning benchmark crafted to expose two frequent failure modes of large vision–language models: language hallucination (answers driven by strong linguistic priors that contradict the image) and visual illusion (misleading visual features that produce confident yet wrong responses). Comprising carefully designed examples that remain challenging for GPT-4V and LLaVA-1.5, it enables fine-grained diagnosis of how VLMs over-trust language or under-exploit vision, offering insights for building more faithfully grounded models.

### C.2.2 Text-Oriented Benchmarks

**TextVQA (Singh et al., 2019)** A benchmark designed to evaluate models' ability to read and reason about text embedded in images. Sourced from the Open Images v3 dataset (Krasin et al., 2017), it includes scenes rich in textual content such as signs and packaging. The benchmark emphasizes integration of OCR with visual and linguistic reasoning. We use the validation split for evaluation.

**AI2D (Kembhavi et al., 2016)** A diagram-based question answering benchmark consisting of over 5,000 grade school science diagrams, annotated with more than 150,000 structured labels and ground-truth syntactic parses. It also includes over 15,000 multiple-choice questions aligned with the diagrams, enabling research on visual reasoning and diagram understanding in scientific contexts. We use the test split for evaluation.

**ChartQA (Masry et al., 2022)** A large-scale benchmark designed for question answering over charts, focusing on complex reasoning that involves both visual interpretation and logical or arithmetic operations. It includes 9.6K human-written questions and 23.1K questions generated from chart summaries. Unlike prior template-based benchmarks, ChartQA challenges models to perform multi-step reasoning using both the visual content and underlying data tables of charts, highlighting the need for advanced multimodal understanding. We use the test split for evaluation.

**OCRBench (Liu et al., 2024d)** A comprehensive evaluation benchmark assessing the OCR capabilities of large multimodal models. It comprises 29 datasets across diverse text-related visual tasks, including text recognition, scene text-centric VQA, document-oriented VQA, key information extraction, and handwritten mathematical expression recognition.

### C.2.3 Video Benchmarks

**MLVU (Zhou et al., 2025)** The first large-scale benchmark for long video understanding. It features videos ranging from 3 minutes to 2 hours and spans nine tasks covering holistic, single-detail, and multi-detail understanding. Both multiple-choice and open-ended questions are included. We report performance using the M-Avg metric.

**MVBench (Li et al., 2023a)** A benchmark tailored for evaluating temporal reasoning in video comprehension. It includes 20 carefully designed tasks requiring dynamic understanding across multiple frames. MVBench introduces a static-to-dynamic transformation approach to systematically test temporal understanding. We use the test split for evaluation.

**LongVideoBench (Wu et al., 2024)** A large-scale benchmark for evaluating understanding of long-form videos. It consists of 3,763 videos (up to 1 hour long), each accompanied by subtitles and 6,678 human-written multiple-choice questions across 17 categories. A key feature is the referring reasoning task, where questions target specific video segments. We use the validation split for evaluation.

**Video-MME (Fu et al., 2025)** A comprehensive video benchmark featuring 900 expert-curated videos spanning 256 hours across six primary domains and 30 subfields. Videos range from 11 seconds to 1 hour in duration and include video, audio, and subtitles (not used during evaluation). It provides 2,700 expert-annotated QA pairs designed to probe complex temporal and multimodal reasoning abilities.

## C.3 Comparison Methods

### C.3.1 Text-based Methods

**FastV (Chen et al., 2024a)** The first work to identify inefficiencies in visual attention within MLLMs. Based on this observation, FastV proposes a simple, training-free acceleration method: after the second transformer layer, it prunes a portion of visual tokens with the lowest visual-text attention scores. This strategy significantly reduces computational cost during inference without retraining.

**SparseVLM (Zhang et al., 2024c)** Inspired by multi-stage pruning methods such as PyramidDrop, SparseVLM introduces a more fine-grained strategy that incorporates textual guidance. It observes that not all instruction tokens are equally informative for pruning visual tokens. Therefore, it first selects text tokens highly relevant to the visual content as "raters," and uses their attention distribution to guide which visual tokens should be preserved or pruned, resulting in improved model efficiency and accuracy.

### C.3.2 Vision-based Methods

**TRIM (Song et al., 2025)** Pruning based solely on visual input, while ignoring textual context, may lead to suboptimal decisions. TRIM addresses this issue by utilizing CLIP-based similarity. It computes cosine similarities between image tokens (from the visual encoder) and text tokens (from the text encoder), and uses these similarity scores to rank visual tokens by importance. Low-similarity tokens are pruned to accelerate inference without significant performance loss.

**VisionZip (Yang et al., 2025b)** A visual-only token pruning method that analyzes self-attention concentration in the visual encoder. VisionZip first selects dominant tokens with high self-attention weights. Then, it applies clustering on the remaining tokens to extract diverse contextual tokens. The union of dominant and contextual tokens is passed to the language model, aiming to preserve both saliency and diversity of visual content.

### C.3.3 Similarity-based Methods

**DivPrune (Alvar et al., 2025)** DivPrune reformulates token pruning as a Maximum Minimum Distance Problem (MMDP), aiming to select the most diverse subset of visual tokens. Rather than relying solely on attention or similarity scores, it explicitly maximizes the minimum pairwise distance among retained tokens, ensuring the selected tokens cover a broad semantic space. This diversity-preserving strategy leads to robust performance under extreme token reduction.

Table D.1: **Performance comparisons on InternVL3-8B Zhu et al. (2025) across 10 image understanding benchmarks**. The best results in each setting are **bolded**, and the second-best are underlined.

| Method | AI2D | TextVQA | ChartQA | OCRBench | HallBench | MME | MMB-EN | MMB-CN | Acc. | Average |
|---|---|---|---|---|---|---|---|---|---|---|
| *Upper Bound, All 1280 Tokens* (**100**%) | | | | | | | | | | |
| InternVL3-8B | 85.28 | 81.51 | 85.07 | 853 | 50.02 | 2393.22 | 83.82 | 82.54 | 84.22 | 100.0% |
| *Retain 256 Tokens* (↓ **80.0**%) | | | | | | | | | | |
| FastV | 82.21 | 74.34 | 70.58 | 632 | 48.45 | 2348,31 | 83.36 | 82.04 | 77.98 | 92.6% |
| DivPrune | 80.88 | 64.70 | 57.51 | 477 | 38.63 | 2249.17 | 80.63 | 80.27 | 70.41 | 82.80% |
| **Script** | 82.87 | 75.97 | 72.20 | 640 | 48.98 | 2334.22 | 83.45 | 81.57 | **78.35** | **93.03%** |
| *Retain 128 Tokens* (↓ **90.0**%) | | | | | | | | | | |
| FastV | 77.35 | 63.55 | 46.82 | 426 | 42.75 | 2250.31 | 81.09 | 80.20 | 68.44 | 80.9% |
| DivPrune | 76.45 | 55.44 | 42.58 | 378 | 37.57 | 2166.22 | 78.49 | 77.55 | 64.31 | 75.75% |
| **Script** | 79.88 | 67.65 | 50.78 | 471 | 44.46 | 2282.98 | 82.12 | 80.53 | **70.98** | **84.29%** |

## C.4 Implementation Details

For image-based benchmarks, we adopt the official implementation of LLaVA[4], loading the released checkpoints (e.g., LLaVA-1.5 and LLaVA-NeXT) and following the default preprocessing pipeline for image resizing (336×336 or 672×672), tokenization, and prompt formatting. All evaluations are conducted in a zero-shot setting unless otherwise specified. For video-based benchmarks, we instead use the official implementation of LLaVA-NeXT[5], which also supports LLaVA-Video. Videos are processed by uniformly sampling 64 frames, resizing each to 384×384, and pooling visual tokens as described in the original LLaVA-Video paper. For evaluation, we further employ the `lmms-eval` toolkit[6], which standardizes metric computation and dataset loading for long-form video understanding.

By default, Script in GSP is configured with $\tau = 0.3$ and $\gamma = 5$. All experiments are run on NVIDIA H100 GPUs (80GB) with bfloat16 precision. Moreover, to ensure fair comparison across models and pruning strategies, we keep inference batch size and decoding settings (temperature $= 0.2$, top-$k = 1$) consistent. Finally, each benchmark is evaluated 3 times, and we report the average as the final score.

## C.5 Licenses

Table C.1 summarizes all benchmarks and software licenses employed in our study. The image understanding benchmarks such as GQA, VQAv2, TextVQA, and VizWiz are released under the CC BY 4.0 license, while others, including ScienceQA, MME, and POPE, adopt the MIT license, and recent large-scale resources such as MMWet, MMBench, and MVBench are distributed under Apache-2.0. For video reasoning, we rely on LongVideoBench and MLVU (Apache-2.0) as well as VideoMME (MIT). On the software side, multimodal models like LLaVA and LLaVA-NEXT are covered by the Llama Community License, whereas InternVL3 is open-sourced under Apache-2.0. We have reviewed the terms of each license, ensured that all datasets and models are publicly available, and strictly limited their usage to non-commercial, academic research purposes. No proprietary or closed-access resources are included, which guarantees reproducibility, transparency, and compliance with community standards.

Table C.1: License information for the scientific artifacts.

| Data Sources | URL | License |
|---|---|---|
| Link | CC BY 4.0 | |
| ScienceQA | Link | MIT |
| VQAv2 | Link | CC BY 4.0 |
| TextVQA | Link | CC BY 4.0 |
| VizWiz | Link | CC BY 4.0 |
| MMVet | Link | Apache-2.0 |
| MMBench | Link | Apache-2.0 |
| MMBench-CN | Link | Apache-2.0 |
| MME | Link | MIT |
| POPE | Link | MIT |
| LongVideoBench | Link | Apache-2.0 |
| VideoMME | Link | MIT |
| MLVU | Link | Apache-2.0 |
| MVBench | Link | Apache-2.0 |
| **Software Code** | **URL** | **License** |
| LLaVA | Link | Llama Community Licence |
| LLaVA-NEXT | Link | Llama Community Licence |
| InternVL3 | Link | Apache-2.0 |

---

[4]https://github.com/haotian-liu/LLaVA
[5]https://github.com/LLaVA-VL/LLaVA-NeXT
[6]https://github.com/EvolvingLMMs-Lab/lmms-eval

Table D.2: **Performance comparisons on LLaVA-1.5-13B Liu et al. (2024a) across 10 image understanding benchmarks**.

| Method | Venue | VQA$^{V2}$ | GQA | VizWiz | SQA$^{IMG}$ | VQA$^{Text}$ | POPE | MME | MMB$^{EN}$ | MMB$^{CN}$ | MMVet | Acc. | Average |
|---|---|---|---|---|---|---|---|---|---|---|---|---|---|
| _Upper Bound, 576 Tokens (100%), 3.817 TFLOPs_ | | | | | | | | | | | | | |
| LLaVA-1.5-13B Liu et al. (2023) | _Nips'23_ | 80.10 | 63.35 | 53.36 | 72.85 | 61.24 | 86.00 | 1531.25 | 68.55 | 63.55 | 36.22 | 66.21 | 100% |
| _Retain 128 Tokens in Average (↓ 77.8%), ~0.833 TFLOPs_ | | | | | | | | | | | | | |
| FastV Chen et al. (2024a) | _ECCV'24_ | 75.53 | 58.13 | 54.46 | 74.32 | 58.76 | 75.55 | 1460.56 | 66.01 | 62.23 | 32.78 | 63.17 | 95.41% |
| TRIM Song et al. (2025) | _COLING'25_ | 76.34 | 59.14 | 49.87 | 72.14 | 55.08 | 86.38 | 1426.49 | 67.11 | 58.14 | 35.13 | 63.06 | 95.24% |
| VisionZip Yang et al. (2025b) | _CVPR'25_ | 76.83 | 57.79 | 52.03 | 73.68 | 58.39 | 82.77 | 1449.42 | 67.14 | 62.95 | 36.01 | 64.01 | 96.67% |
| DivPrune Alvar et al. (2025) | _CVPR'25_ | 77.41 | 59.21 | 53.35 | 72.28 | 58.40 | 86.81 | 1457.97 | 66.13 | 60.74 | 34.41 | 64.16 | 96.90% |
| SparseVLM Zhang et al. (2024c) | _ICML'25_ | 77.61 | 59.46 | 51.74 | 74.23 | 59.93 | 85.02 | 1487.59 | 68.14 | 62.36 | 35.92 | 64.87 | 97.97% |
| **Script (Ours)** | _Proposed_ | 77.87 | 59.27 | 52.49 | 73.29 | 58.45 | 87.31 | 1498.30 | 67.15 | 61.35 | 36.12 | **64.83** | **97.64%** |
| _Retain 64 Tokens in Average (↓ 88.9%), ~0.415 TFLOPs_ | | | | | | | | | | | | | |
| FastV Chen et al. (2024a) | _ECCV'24_ | 65.73 | 51.39 | 53.48 | 73.01 | 53.74 | 56.49 | 1246.54 | 59.12 | 55.18 | 26.89 | 55.78 | 84.24% |
| TRIM Song et al. (2025) | _COLING'25_ | 73.12 | 57.85 | 49.23 | 72.00 | 52.10 | 86.65 | 1406.12 | 65.04 | 52.57 | 27.98 | 60.73 | 91.72% |
| VisionZip Yang et al. (2025b) | _CVPR'25_ | 73.75 | 56.23 | 53.12 | 74.29 | 57.41 | 75.67 | 1379.66 | 64.94 | 61.33 | 33.40 | 61.91 | 93.51% |
| DivPrune Alvar et al. (2025) | _CVPR'25_ | 75.20 | 57.91 | 54.49 | 71.67 | 57.54 | 84.35 | 1454.29 | 64.13 | 59.87 | 29.31 | 62.73 | 94.74% |
| SparseVLM Zhang et al. (2024c) | _ICML'25_ | 73.12 | 55.91 | 52.17 | 72.09 | 57.14 | 77.91 | 1374.35 | 65.12 | 60.23 | 32.94 | 61.62 | 93.07% |
| **Script (Ours)** | _Proposed_ | 76.67 | 59.64 | 53.55 | 72.75 | 57.86 | 87.19 | 1466.98 | 65.85 | 58.72 | 36.20 | **64.20** | **96.96%** |
| _Retain 32 Tokens in Average (↓ 94.5%), ~0.208 TFLOPs_ | | | | | | | | | | | | | |
| FastV Chen et al. (2024a) | _ECCV'24_ | 61.13 | 48.36 | 51.68 | 72.37 | 50.73 | 53.99 | 1198.33 | 54.27 | 53.22 | 23.65 | 52.93 | 79.95% |
| TRIM Song et al. (2025) | _COLING'25_ | 69.83 | 55.67 | 48.58 | 70.64 | 49.65 | 85.85 | 1284.57 | 63.13 | 45.45 | 26.34 | 57.79 | 86.28% |
| VisionZip Yang et al. (2025b) | _CVPR'25_ | 68.40 | 52.71 | 53.09 | 72.79 | 55.20 | 66.85 | 1257.67 | 61.32 | 55.58 | 29.43 | 57.81 | 87.62% |
| DivPrune Alvar et al. (2025) | _CVPR'25_ | 72.00 | 56.20 | 54.55 | 70.89 | 54.76 | 79.12 | 1405.02 | 61.47 | 57.12 | 27.98 | 60.42 | 91.25% |
| SparseVLM Zhang et al. (2024c) | _ICML'25_ | 71.57 | 54.05 | 51.54 | 70.86 | 53.74 | 77.45 | 1327.37 | 62.88 | 58.91 | 28.13 | 59.55 | 89.94% |
| **Script (Ours)** | _Proposed_ | 75.25 | 58.75 | 53.35 | 71.99 | 55.43 | 87.31 | 1421.10 | 63.79 | 56.36 | 30.77 | **62.41** | **94.26%** |

Table D.3: **Performance comparisons on LLaVA-Next-13B Liu et al. (2024a) across 8 image understanding benchmarks**. The best results in each setting are **bolded**, and the second-best are underlined.

| Method | Venue | VQA$^{V2}$ | GQA | VizWiz | SQA$^{IMG}$ | VQA$^{Text}$ | POPE | MME | MMB$^{EN}$ | MMB$^{CN}$ | MMVet | Acc. | Average |
|---|---|---|---|---|---|---|---|---|---|---|---|---|---|
| _Upper Bound, 576 Tokens (100%), 3.817 TFLOPs_ | | | | | | | | | | | | | |
| LLaVA-NeXT-13B Liu et al. (2024a) | _Nips'23_ | 82.30 | 64.35 | 59.15 | 73.15 | 63.20 | 85.20 | 1539.50 | 68.55 | 61.05 | 45.02 | 67.91 | 100% |
| _Retain 640 Tokens in Average (↓ 77.8%), ~4.627 TFLOPs_ | | | | | | | | | | | | | |
| FastV Chen et al. (2024a) | _ECCV'24_ | 79.34 | 60.89 | 56.44 | 71.75 | 60.74 | 80.24 | 1536.7 | 65.25 | 59.39 | 43.68 | 65.45 | 96.38% |
| TRIM Song et al. (2025) | _COLING'25_ | 79.34 | 63.19 | 54.31 | 71.42 | 57.64 | 87.31 | 1543.36 | 68.75 | 61.21 | 42.34 | 66.26 | 97.57% |
| VisionZip Yang et al. (2025b) | _CVPR'25_ | 79.37 | 62.79 | 56.12 | 70.83 | 61.91 | 85.83 | 1529.22 | 68.31 | 62.61 | 46.98 | 67.13 | 98.84% |
| DivPrune Alvar et al. (2025) | _CVPR'25_ | 80.34 | 63.54 | 56.73 | 72.12 | 59.42 | 86.44 | 1531.41 | 67.51 | 62.59 | 39.10 | 66.42 | 97.81% |
| SparseVLM Zhang et al. (2024c) | _ICML'25_ | 79.49 | 62.74 | 57.55 | 72.35 | 62.48 | 85.56 | 1573.74 | 68.85 | 64.10 | 41.35 | 67.31 | 99.11% |
| **Script (Ours)** | _Proposed_ | 81.14 | 64.22 | 57.02 | 72.08 | 61.40 | 87.31 | 1552.61 | 68.93 | 61.91 | 47.13 | **67.87** | **99.95%** |
| _Retain 320 Tokens in Average (↓ 88.9%), ~2.314 TFLOPs_ | | | | | | | | | | | | | |
| FastV Chen et al. (2024a) | _ECCV'24_ | 66.98 | 54.36 | 53.33 | 70.11 | 55.34 | 64.10 | 1288.0 | 59.48 | 54.14 | 30.52 | 57.27 | 84.34% |
| TRIM Song et al. (2025) | _COLING'25_ | 75.94 | 61.13 | 52.12 | 69.59 | 52.18 | 87.42 | 1484.6 | 67.53 | 57.34 | 33.41 | 63.09 | 92.90% |
| VisionZip Yang et al. (2025b) | _CVPR'25_ | 76.18 | 60.37 | 54.84 | 70.12 | 60.17 | 82.33 | 1417.13 | 66.45 | 62.53 | 41.21 | 64.50 | 94.96% |
| DivPrune Alvar et al. (2025) | _CVPR'25_ | 78.13 | 61.28 | 55.10 | 72.39 | 57.46 | 85.32 | 1465.01 | 65.91 | 61.59 | 39.42 | 64.98 | 95.68% |
| SparseVLM Zhang et al. (2024c) | _ICML'25_ | 76.75 | 60.91 | 54.75 | 70.94 | 60.01 | 81.35 | 1491.46 | 68.10 | 63.45 | 39.36 | 65.05 | 95.78% |
| **Script (Ours)** | _Proposed_ | 79.64 | 63.11 | 55.31 | 71.46 | 58.74 | 87.63 | 1501.15 | 66.73 | 61.48 | 42.32 | **66.16** | **97.42%** |
| _Retain 160 Tokens in Average (↓ 94.4%), ~1.156 TFLOPs_ | | | | | | | | | | | | | |
| FastV Chen et al. (2024a) | _ECCV'24_ | 62.67 | 50.79 | 52.03 | 69.11 | 51.74 | 62.95 | 1229.18 | 56.48 | 53.16 | 27.36 | 54.77 | 80.66% |
| TRIM Song et al. (2025) | _COLING'25_ | 72.15 | 58.93 | 51.12 | 69.19 | 49.12 | 87.00 | 1392.30 | 65.75 | 51.65 | 27.68 | 60.21 | 88.66% |
| VisionZip Yang et al. (2025b) | _CVPR'25_ | 72.45 | 57.81 | 52.35 | 69.47 | 58.69 | 76.38 | 1393.49 | 64.85 | 60.01 | 35.39 | 61.70 | 90.86% |
| DivPrune Alvar et al. (2025) | _CVPR'25_ | 75.64 | 60.30 | 53.15 | 71.44 | 56.43 | 81.29 | 1436.17 | 65.19 | 60.59 | 37.34 | 63.31 | 93.22% |
| SparseVLM Zhang et al. (2024c) | _ICML'25_ | 74.62 | 59.79 | 52.38 | 69.89 | 55.89 | 80.92 | 1429.44 | 65.19 | 59.17 | 36.48 | 62.58 | 92.15% |
| **Script (Ours)** | _Proposed_ | 77.85 | 62.32 | 53.19 | 71.37 | 56.47 | 88.83 | 1476.99 | 65.49 | 59.91 | 40.34 | **64.96** | **95.65%** |

# D Additional Experimental Results

## D.1 Script for advanced open-source MLLM

In addition to LLaVA, we further apply Script to one of the most advanced open-source MLLMs to date, InternVL3. The results are shown in Table D.1. Here, we fix the input resolution to 896×896, yielding 1,280 visual tokens. Notably, unlike its performance on the LLaVA series, DivPrune exhibits a significant performance drop on InternVL3, as it does not account for the relevance to user instructions during pruning. In contrast, our Script jointly considers both diversity and relevance, consistently achieving the best performance across different reduction ratios. Specifically, even when 90% of the visual tokens are removed, our method retains 83.9% of the original performance, 3% higher than the second-best FastV, demonstrating its effectiveness and adaptability in advanced MLLM architectures.

## D.2 Script for Large Parameters of Model

To evaluate the effectiveness of our proposed method on larger language models, we apply Script to two models equipped with 13B LLMs: LLaVA-1.5-13B and LLaVA-NeXT-13B. The results are presented in

Table D.4: Hyperparameter sensitivity analysis of Scripts on LLaVA-1.5-7B with 64 tokens retained. The hyperparameters include the graph threshold $\tau$ (0.1, 0.3, 0.5, 0.7, 0.9), scaling factor $\gamma$ (1, 10, 50, 100), and kernel choices ($S$ vs. $S'$). **Bold** is the default setting (Hyperparameter).

| Hyperparameter | Benchamark | | | | | | | | | | Average | Relative |
|---|---|---|---|---|---|---|---|---|---|---|---|---|
| | VQA$^{V2}$ | GQA | VizWiz | SQA$^{IMG}$ | VQA$^{Text}$ | POPE | MME | MMB$^{EN}$ | MMB$^{CN}$ | MMVet | | |
| *Upper Bound, 576 Tokens (100%), 3.817 TFLOPs* | | | | | | | | | | | | |
| | 61.94 | 64.09 | 58.10 | 1507.06 | 86.96 | 69.41 | 78.50 | 58.20 | 50.32 | 31.82 | 63.47 | 100% |
| $\tau$ (graph threshold, **0.1, 0.3, 0.5, 0.7, 0.9**) | | | | | | | | | | | | |
| 0.1 | 59.07 | 61.30 | 51.43 | **1421.74** | 84.76 | 66.88 | 73.98 | **55.31** | **54.44** | 28.76 | 60.71 | 95.64% |
| **0.3** | 59.28 | **61.90** | **52.93** | 1412.08 | **86.95** | **68.65** | 75.08 | 55.20 | 54.31 | **29.96** | **61.49** | 96.88% |
| 0.5 | **59.33** | **61.90** | 52.71 | 1409.31 | 85.16 | 68.00 | 74.28 | 55.10 | 54.07 | 29.06 | 61.01 | 96.12% |
| 0.7 | 58.41 | 60.92 | 52.48 | 1394.89 | 84.05 | 67.45 | 74.72 | 54.99 | 53.98 | 28.77 | 60.56 | 95.42% |
| 0.9 | 58.77 | 60.52 | 51.94 | 1377.28 | 85.74 | 67.34 | 74.11 | 53.82 | 53.99 | 29.41 | 60.45 | 95.25% |
| $\gamma$ (Scaling factor) | | | | | | | | | | | | |
| 1 | 59.11 | 61.50 | 52.77 | 1399.22 | 86.80 | 66.40 | 74.55 | 53.90 | 53.50 | 29.87 | 60.83 | 95.85% |
| **5** | 59.28 | **61.90** | 52.93 | 1412.08 | **86.95** | **68.65** | 75.08 | 55.20 | **54.31** | 29.96 | **61.49** | 96.88% |
| 10 | 59.30 | 61.80 | **52.95** | **1415.20** | 86.75 | 67.32 | 74.68 | **55.70** | 54.10 | 29.20 | 61.25 | 96.51% |
| 50 | 59.35 | **61.90** | 52.90 | 1407.75 | 86.60 | 67.75 | 73.88 | 54.75 | 54.20 | 29.55 | 61.13 | 96.31% |
| 100 | **59.38** | **61.90** | 52.90 | 1412.02 | 86.90 | 68.20 | 74.35 | 54.66 | 54.30 | **30.30** | 61.35 | 96.66% |
| Kernel Choices ($S$ **vs.** $S'$) | | | | | | | | | | | | |
| $S$ | **59.58** | 60.30 | 51.75 | 1399.20 | 85.40 | 66.25 | **75.30** | 54.40 | 53.11 | 28.77 | 60.47 | 95.28% |
| $S'$ | 59.28 | **61.90** | **52.93** | **1412.08** | **86.95** | **68.65** | 75.08 | **55.20** | **54.31** | **29.96** | **61.49** | 96.88% |

Table D.2 and Table D.3. The larger language models lead to significant performance improvements and also make MLLMs less sensitive to visual token pruning. Among various pruning strategies, text-attention-based methods benefit the most from scaling up the language model, indicating that a larger LLM brings more accurate attention. Across different types of pruning methods, Script consistently outperforms all other approaches under various reduction ratios. With 77.8% of visual tokens removed, our method retains 97.64% and 99.95% of the original performance on LLaVA-1.5-13B and LLaVA-NeXT-13B, respectively, demonstrating its effectiveness on larger language models.

## D.3 Hyperparameter sensitivity & guidance.

To better understand the robustness of Scripts, we conduct a hyperparameter sensitivity analysis under the setting where 64 tokens are retained. Specifically, we vary the graph threshold $\tau$ , the scaling factor $\gamma$ in Eq. 9, and kernel choices ($S$ vs. $S'$). As shown in Table D.4, our method demonstrates remarkable robustness to the choice of threshold: across $\tau \in [0.1, 0.9]$, the average relative performance consistently remains above 95%, indicating minimal sensitivity to this hyperparameter. Moreover, performance varies smoothly and predictably with $\tau$, without instability or abrupt drops, further confirming the stability and reliability of our approach under different settings. In practice, extremely low (0.1) or high (0.9) values lead to slightly worse performance, while moderate settings achieve a better trade-off. Among them, $\tau = 0.3$ yields the strongest overall results. For the scaling factor $\gamma$, all values produce competitive outcomes, but $\gamma = 5$ achieves the most consistent improvements across benchmarks. In terms of kernel design, $S'$ consistently outperforms $S$, suggesting that it better captures redundancy patterns among tokens. In the comparison between $S$ and $S'$, the former represents the redundancy of tokens informed by visual similarity, while the latter incorporates both visual redundancy and query relevance. Based on these observations, we use $\tau = 0.3$, $\gamma = 5$, and $S'$ as the default configuration.

It is worth noting that in Script, the parameter $k$ in DPP is not treated as a tunable hyperparameter but as a direct control of the number of retained tokens. This is conceptually equivalent to the pruning ratio $p$ in GSP, with the correspondence:

$$k = (1 - p)\, n, \tag{D.1}$$

where $n$ is the number of original tokens. In other words, while $p$ specifies the proportion of redundant tokens to be discarded, $k$ explicitly determines the number of tokens to be preserved by ranking redundancy from low to high. Detailed results under different $k$ settings are reported in Section 5 and Appendix D.

Table D.5: Intersection dynamics (GSP ∩ QCSP)

| Benchamark | | | | | | | | | | Average |
|---|---|---|---|---|---|---|---|---|---|---|
| VQA$^{\text{V2}}$ | GQA | VizWiz | SQA$^{\text{IMG}}$ | VQA$^{\text{Text}}$ | POPE | MME | MMB$^{\text{EN}}$ | MMB$^{\text{CN}}$ | MMVet | |
| *Retain 192 Tokens in Average ($\downarrow$ 66.7%), $\sim$1.253TFLOPs* | | | | | | | | | | |
| 45.69 | 45.48 | 54.40 | 47.28 | 48.72 | 44.53 | 49.32 | 52.44 | 48.21 | 49.08 | 48.52 |
| *Retain 128 Tokens in Average ($\downarrow$ 77.8%), $\sim$0.833 TFLOPs* | | | | | | | | | | |
| 33.29 | 33.78 | 40.89 | 33.68 | 36.75 | 31.55 | 31.50 | 42.12 | 37.51 | 36.26 | 35.73 |
| *Retain 64 Tokens in Average ($\downarrow$ 88.9%), $\sim$0.415 TFLOPs* | | | | | | | | | | |
| 21.89 | 23.05 | 26.25 | 21.32 | 26.74 | 19.27 | 20.81 | 28.71 | 23.42 | 24.58 | 23.61 |
| *Retain 32 Tokens in Average ($\downarrow$ 94.5%), $\sim$0.208 TFLOPs* | | | | | | | | | | |
| 15.75 | 16.45 | 17.49 | 15.53 | 19.38 | 11.95 | 14.85 | 15.51 | 18.83 | 18.44 | 16.42 |
| *Retain 16 Tokens in Average ($\downarrow$ 97.3%), $\sim$0.103 TFLOPs* | | | | | | | | | | |
| 10.73 | 11.16 | 10.83 | 11.77 | 11.57 | 6.82 | 9.84 | 11.72 | 12.62 | 12.94 | 11.00 |

## D.4 Overlap analysis between GSP and QCSP.

GSP and QCSP are two complementary modules that operate from different perspectives and together determine which tokens are retained. Table D.5 quantifies the overlap size between the two modules under different retention budgets.

Overall, the overlap ratio decreases steadily as the pruning ratio increases. This trend is intuitive: when fewer tokens are allowed to remain, it becomes more difficult for both modules to consistently select the same tokens, given their distinct selection criteria. For example, retaining 192 tokens leads to an average overlap of 66.7%, while retaining only 16 tokens reduces the overlap to 97.3%, indicating that QCSP dominates token retention under stricter pruning.

Another noteworthy observation is that the overlap never reaches 100%, which necessitates the use of QCSP as a fallback mechanism to ensure the desired number of tokens is met. As the pruning becomes more aggressive, QCSP contributes proportionally more to the final selection. At the same time, our ablation studies confirm that GSP provides complementary benefits by capturing structural redundancy patterns that QCSP alone may overlook.

These results highlight the importance of the cooperative design: GSP guides pruning with a global structural view, while QCSP ensures query relevance and satisfies token budget constraints. Their interaction achieves a balance between efficiency and accuracy, with QCSP gradually taking on a stronger role as the pruning ratio increases.

## D.5 Statistical Significance Experiments

To assess the robustness and statistical reliability of our method, we therefore conduct 20 independent runs using LLaVA-Next-7B under identical experimental settings with varying random seeds, as shown in Table D.6. All statistical comparisons are made consistently against the Script. To determine whether Script is statistically better than other methods, we accordingly perform paired one-sided $t$-tests and Wilcoxon signed-rank tests. We explicitly evaluate the null hypothesis $H_0$: Script performs the same as the others, against the alternative hypothesis $H_1$: Script performs better. Since the resulting

Table D.6: Statistical comparison between our method and other baselines on MME benchmark with ratain 160 tokens. **StdDev** denotes standard deviation. Reported $p$-values correspond to the paired one-sided $t$-test and Wilcoxon signed-rank test. All experiments are set with the significance level of $\alpha = 5\%$.

| Method | Mean (%) | StdDev | $t$-test $p$ | Wilcoxon $p$ |
|---|---|---|---|---|
| FastV | 1076.81 | 2.62 | $1.01 \times 10^{-43}$ | $9.54 \times 10^{-7}$ |
| TRIM | 1316.74 | 3.41 | $8.96 \times 10^{-35}$ | $9.54 \times 10^{-7}$ |
| VisionZip | 1349.28 | 2.34 | $8.49 \times 10^{-36}$ | $9.54 \times 10^{-7}$ |
| DivPrune | 1347.72 | 2.04 | $1.90 \times 10^{-36}$ | $9.54 \times 10^{-7}$ |
| SparseVLM | 1369.62 | 2.39 | $7.58 \times 10^{-35}$ | $9.54 \times 10^{-7}$ |
| **Script(Ours)** | 1487.98 | 0.94 | 1.00 | 1.00 |

$p$-values are significantly lower than the significance threshold of $\alpha = 0.05$, we can confidently reject the null hypothesis $H_0$. This suggests that the observed performance difference is statistically significant.

# E    Additional Visualization Results

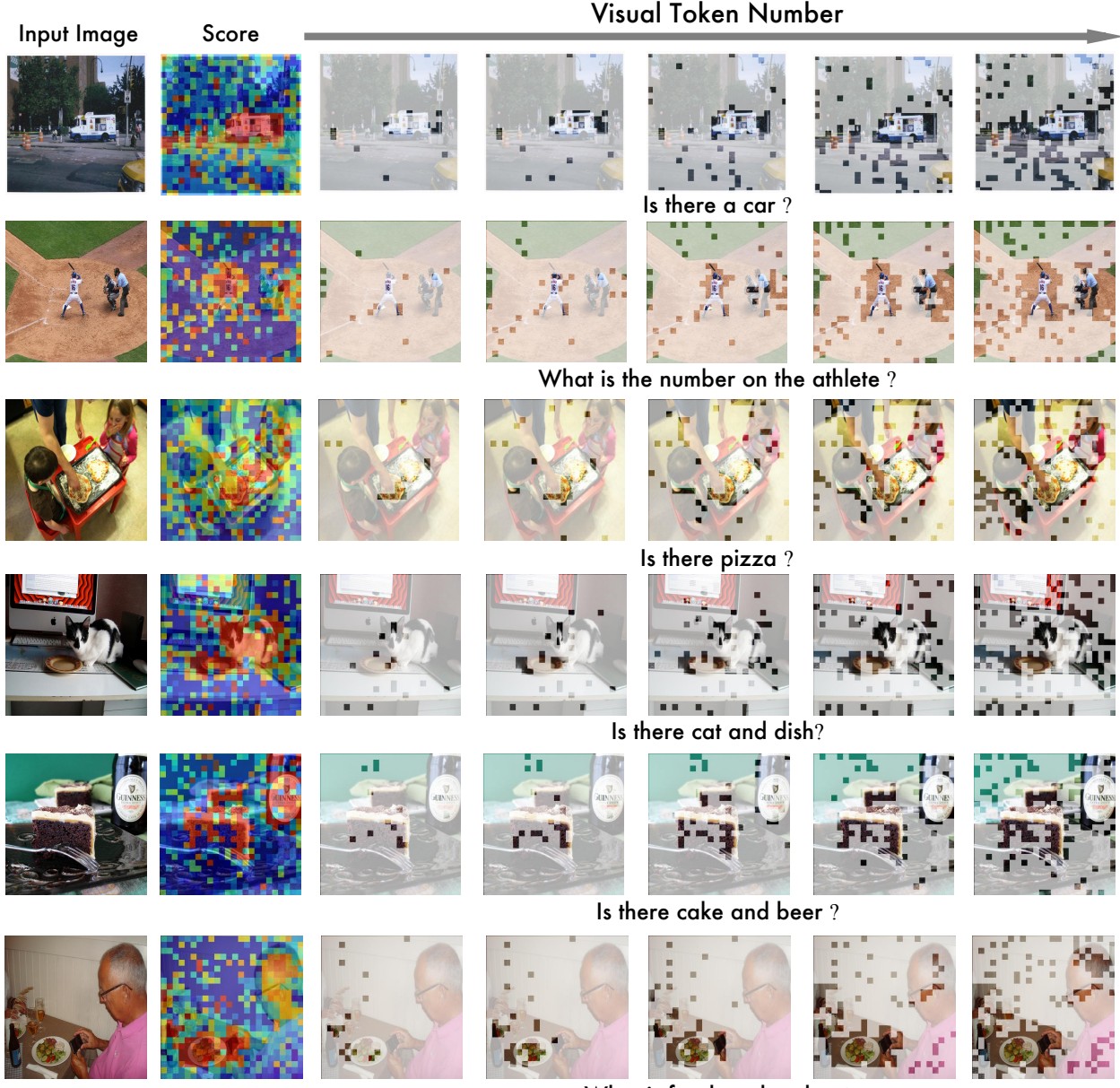

Figure E.1: **Visualizations of relevance scores and retained tokens.** Each visualization illustrates the spatial attention allocated by models to regions corresponding to various textual instructions, demonstrating the capacity of pre-trained multimodal models to identify and focus on task-specific visual elements.

## E.1    Case study

In this section, we provide additional visualizations that comprehensively illustrate the relevance scores and retained visual tokens as shown in Figures E.1 and E.2. These visualizations further emphasize the strengths of models utilizing language-image pre-training, showcasing their enhanced capability to align textual instructions accurately with the corresponding visual regions. Specifically, the visualizations depict clear and intuitive correspondences: for instance, when instructions involve detecting the presence of specific objects or counting individuals or animals, the relevance maps highlight pertinent regions directly associated with

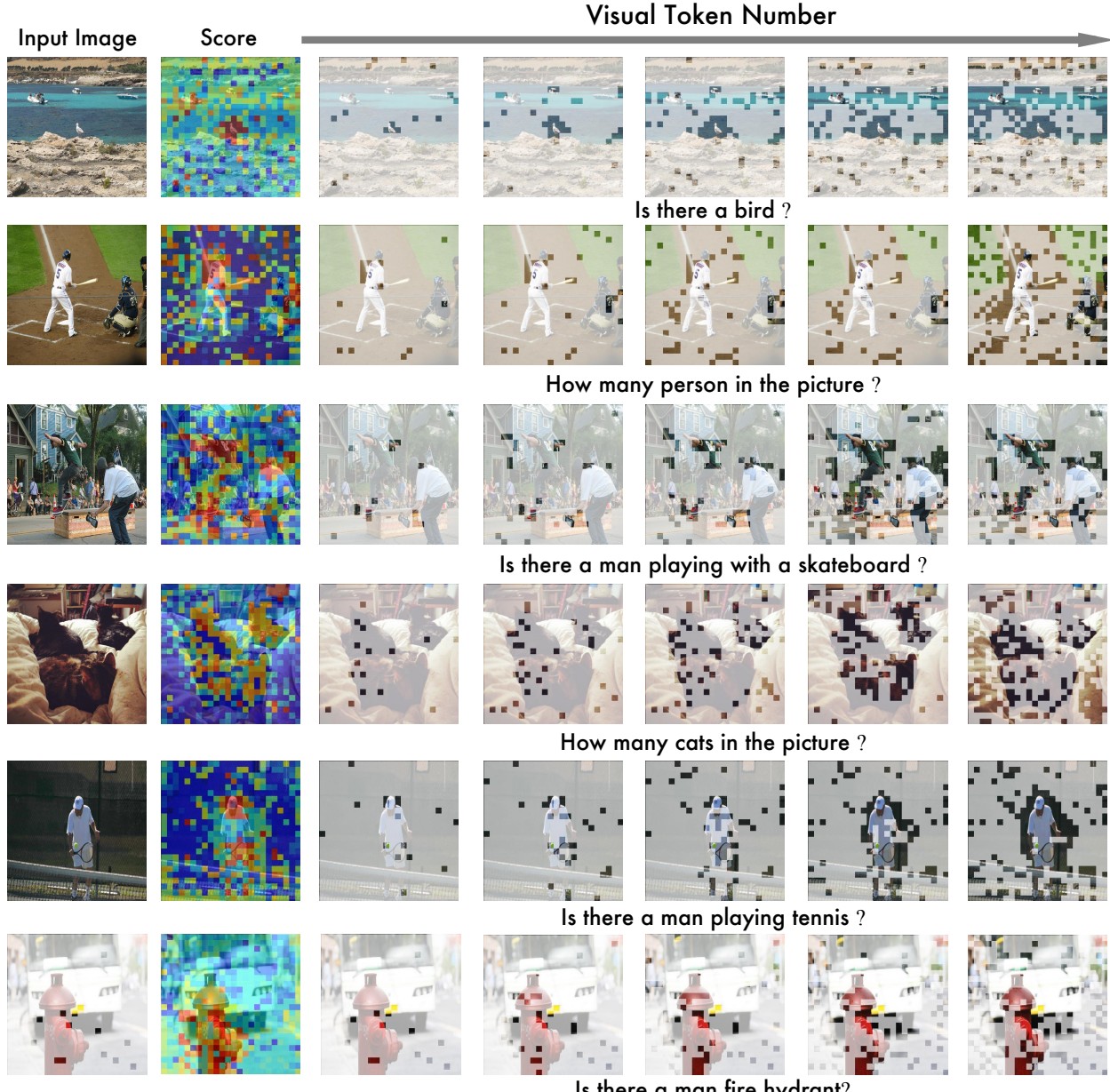

Figure E.2: **Visualizations of relevance scores and retained tokens.** Each visualization illustrates the spatial attention allocated by models to regions corresponding to various textual instructions, demonstrating the capacity of pre-trained multimodal models to identify and focus on task-specific visual elements.

these queries. Such precise spatial attention enables effective and efficient visual token pruning, significantly reducing redundancy by retaining only informative regions. This targeted retention not only improves computational efficiency but also boosts interpretability by transparently displaying the reasoning processes of multimodal large language models. Overall, these visualizations underscore the robustness and adaptability of pre-trained models in dynamically identifying and concentrating on task-specific visual details essential for accurate multimodal understanding.

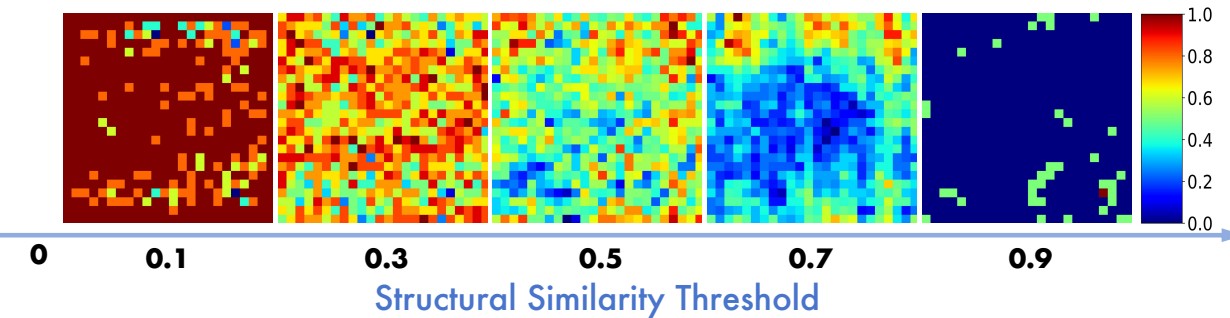

Figure E.3: **Visualization of structural redundancy under different thresholds.** We show patch-level redundancy maps computed from CLIP-ViT features on images from the COCO dataset. Redundancy is defined as the average cosine similarity between each patch and its spatial neighbors. **Red** regions indicate high structural redundancy, while **blue** regions are more distinctive. As the threshold increases (from left to right: 0.1 to 0.9), the resulting selection becomes increasingly sparse, focusing on structurally salient regions.

### E.2 Redundancy Visualization Across Thresholds

To better understand how structural redundancy varies across visual patches, we visualize the *redundancy score maps* under multiple thresholds ($\tau = 0.1, 0.3, 0.5, 0.7, 0.9$), as shown in Fig. E.3. For each image, a patch-wise redundancy score is computed by averaging the cosine similarity of each patch with its 8-connected spatial neighbors. Binary masks are then obtained by applying different thresholds $\tau$ to these scores.

From these visualizations, several detailed observations emerge:

**1. Very low thresholds saturate the map.** At $\tau = 0.1$, almost every patch is marked redundant, producing a mask that is nearly saturated. This indicates that even weak local correlations are enough to pass the threshold, causing both background and structured regions to be flagged. Such masks preserve nearly all information but fail to distinguish between repetitive and unique content, providing little benefit for redundancy reduction.

**2. Progressive sparsification with increasing thresholds.** As $\tau$ increases to 0.3 and 0.5, the maps become less dense and begin to highlight differences between structurally homogeneous regions and structurally diverse ones. Patches in flat or repetitive textures remain marked as redundant, while more irregular patches start to drop out. This progression shows how the threshold acts as a filter, gradually shifting the representation from "inclusive" to "selective."

**3. Emergence of discriminative structures at mid thresholds.** Around $\tau = 0.5$, the contrast between redundant and non-redundant patches becomes clearer. Large homogeneous regions are consistently retained as redundant, but edges, boundaries, and irregular textures are less frequently included. This suggests that mid thresholds achieve a balance: they still capture redundancy for compression purposes, yet they allow distinctive features to remain available for recognition or further processing.

**4. Strong filtering at high thresholds.** At $\tau = 0.7$ and especially $\tau = 0.9$, the redundancy maps become extremely sparse. Only the most distinctive patches, which sharply differ from their neighbors, survive as non-redundant. This makes the mask highly selective, isolating unique structures while discarding most background and repeated patterns. However, such aggressiveness may also risk losing subtle contextual cues that are semantically useful.

**5. Spatial clustering of redundancy.** Across all thresholds, redundant patches tend not to appear as isolated points but rather form spatially contiguous clusters. This behavior indicates that redundancy is inherently a local phenomenon, strongly driven by the short-range similarity between neighboring patches. It also reinforces the idea that large uniform areas will consistently be flagged together, rather than in scattered fragments.

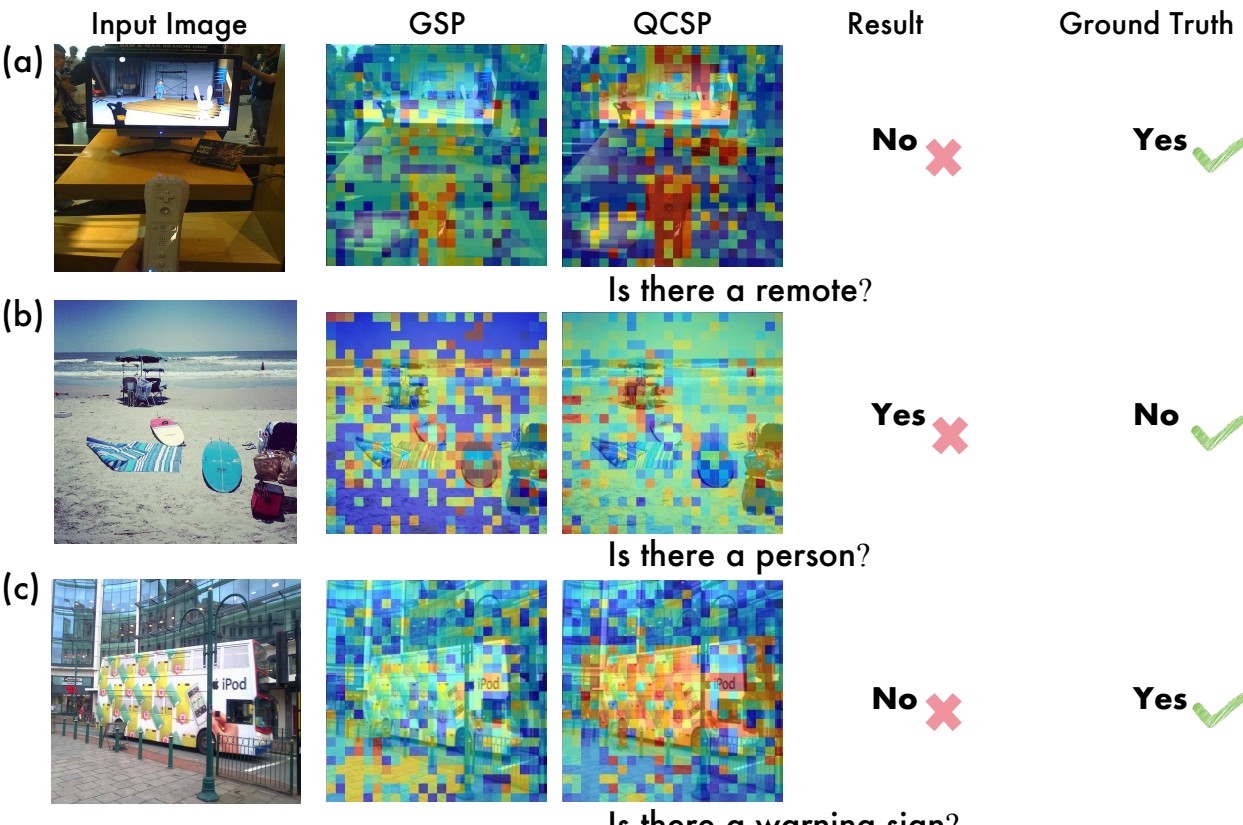

Figure E.4: **Error case analysis.** Representative failure patterns when pre-trained multimodal models process task-specific queries, comparing query, Graph-Structured Pruning (GSP), Query-Conditioned Semantic Pruning (QCSP), model prediction, and ground truth. (a) *Model capability:* success when textual guidance suffices (GSP correct, QCSP correct). (b) *Text–visual misalignment:* errors from limited CLIP encoder alignment (GSP correct, QCSP incorrect). (c) *Visual feature deficiency:* failures due to inadequate CLIP visual representations, leading to incorrect grounding and relevance (GSP incorrect, QCSP incorrect).

**Summary of trade-offs.** Threshold selection directly governs the density and informativeness of the redundancy maps. Low thresholds preserve nearly all content but provide little compression, while high thresholds isolate only the most distinctive patches but risk losing contextual information. Mid-level thresholds (e.g., $\tau \approx 0.5$) offer a favorable compromise, retaining structural diversity while still filtering out redundant clusters. This motivates our pruning strategy, which leverages structural redundancy to maintain discriminative information while reducing redundancy.

### E.3  Error Analysis

The case study in Figures E.1 and E.2 has provided an excellent visualization of a success case. To provide a more balanced perspective, it is equally important to include and discuss failure cases. These highlight the inherent limitations of current pruning strategies and offer insight into potential directions for improvement. As shown in Figure E.4, we highlight three illustrative types of errors, though in reality these categories are not mutually exclusive and some overlap may exist.

(a) **Model capability error.** This case illustrates the inherent limitations of the underlying LLM. Both GSP and QCSP assign relatively high retention scores to the visual token corresponding to the remote control. However, the final prediction is still incorrect. This suggests that even when

query-relevant tokens are preserved, reasoning failures may occur due to the LLM's limited capacity in associating visual evidence with the query semantics.

(b) **Text–visual misalignment.** Here, QCSP assigns higher scores to tokens related to the chairs under the parasol, while GSP distributes attention in a scattered manner. The model incorrectly concludes that a person is present. This reflects a semantic drift issue, where token relevance scoring does not fully capture the nuanced mapping between textual concepts ("person") and visual regions ("chair"), leading to confusion in visually similar contexts. Such errors highlight the need for stronger cross-modal grounding mechanisms that can better align text queries with the correct visual entities.

(c) **Compounded pruning error.** In this example, both GSP and QCSP assign misleading values to tokens, causing the warning sign to be pruned or overlooked. As a result, the system outputs an incorrect response. Unlike case (a), the error here stems from cumulative misjudgments at the pruning stage rather than downstream reasoning. This indicates that pruning errors can directly suppress crucial evidence, creating an information bottleneck that the LLM cannot recover from.

In summary, these failure cases expose three distinct but complementary challenges: (i) intrinsic reasoning limits of the base LLM, (ii) misalignment between text semantics and visual grounding, and (iii) error propagation from aggregated pruning strategies. Together, they emphasize that while Script substantially improves efficiency and often preserves accuracy, there remain scenarios where query-specific signals are underutilized or entirely lost. Future work could address these issues by incorporating adaptive confidence calibration, multimodal consistency checks, or dynamic pruning thresholds that adjust based on query difficulty.

## F   Limitations and Future Work.

One key limitation of our work is that the proposed pruning method requires direct access to encoded visual tokens during inference, which restricts its applicability to open-source Multimodal Large Language Models. By contrast, widely used proprietary systems such as ChatGPT, Gemini, and Claude are closed black boxes where intermediate visual features are inaccessible. Although these models also incur high computational costs for visual reasoning, our method cannot be deployed in such settings.

Another limitation is that our evaluation is limited to vision–language inputs, assuming queries and context are restricted to text and visual tokens. We do not consider other modalities such as audio, depth, or IMU signals. Multimodal tasks like audio-grounded dialogue, audio-visual question answering, or egocentric video reasoning introduce challenges beyond vision–language settings. For instance, audio streams are temporally dense and require alignment between speech content, prosody, and visual context, while depth and IMU data carry fine-grained geometric and motion cues that evolve continuously over time. Efficient pruning in such scenarios would require not only modality-specific encoders but also redundancy metrics that capture temporal correlations (e.g., repeated acoustic frames or redundant motion patterns) rather than purely spatial visual redundancy. While exploring these directions would broaden the scope of our approach, they fall outside the primary focus of this work and are left for future investigation.

Moreover, while our method is compatible with advanced open-source MLLMs such as Qwen2.5-VL and InternVL3, we observe that these models are more sensitive to visual token pruning compared to the LLaVA series. Under the same pruning ratio, they exhibit more significant performance degradation. This likely stems from the fact that such architectures already incorporate internal visual token compression techniques, such as pixel unshuffle or token merging, which reduce redundancy prior to pruning. As a result, additional pruning may lead to excessive information loss. Adapting pruning strategies to better suit these optimized architectures, for example, through pruning-aware training, modality-specific heuristics, or model-adaptive redundancy metrics, remains an important direction for future research.

Finally, our method requires manually selecting the threshold $\tau$. Although our sensitivity analysis (Table D.4) shows relatively stable performance across a wide range of values and we provide default recommendations, the threshold remains fixed rather than adaptively determined. Future work could explore automated or learning-based strategies for setting $\tau$, such as validation-driven calibration, distribution-aware rules (e.g.,

Otsu or quantile-based selection on similarity scores), or meta-learning schemes that condition $\tau$ on image/-query characteristics. Such strategies may further improve robustness under domain shift and heterogeneous token redundancy profiles. We believe addressing this limitation will enhance the practicality and generalizability of pruning in multimodal systems.

## G    Broader Impacts

Multimodal large language models have demonstrated remarkable success across a wide range of domains, including education, accessibility, robotics, and content creation. Despite their capabilities, these models often incur high inference costs, particularly when processing high-resolution images or extended video sequences. Such computational demands pose substantial challenges for real-world deployment.

In this work, we introduce a simple yet effective visual token pruning strategy that enhances inference efficiency without requiring any additional training. By selectively removing redundant visual inputs, our method significantly reduces computational overhead and deployment costs. This improvement facilitates the deployment of MLLMs on resource-constrained platforms, such as mobile devices and edge computing environments, thereby promoting broader accessibility and scalability in practical applications.

However, it is important to note that improving computational efficiency does not inherently address the ethical challenges associated with MLLMs. Our method does not mitigate risks related to potential misuse, such as the generation of harmful content or the dissemination of misinformation. As such, continued research and policy development are essential to ensure the responsible and safe use of increasingly efficient multimodal language models.

