# OpenReview forum: "Script: Graph-Structured and Query-Conditioned Semantic Token Pruning for Multimodal Large Language Models"
_TMLR — Accepted by TMLR_

### Review · Reviewer_w6Ja · 2025-08-26

**Summary Of Contributions:**

The work tackles the issue of visual token explosion in MLLMs, presenting a practical solution for this. The dual pruning strategy, balances efficiency with semantic preservation. Being plug-and-play with no retraining requirement makes the method widely applicable across different MLLMs. Extensive experiments on fourteen benchmarks, covering both image and video understanding, demonstrate strong and consistent results. The reported gains (up to 6.8× prefill speedup, 10× FLOP reduction, and retention of nearly 97% performance) are nice and show practical significance.
The abstract communicates motivation, approach, and contributions clearly and succinctly.

The integration of graph-structured redundancy reduction with DPP-based query-aware pruning is original and well-justified. By modeling token-query interactions, Script avoids the pitfalls of prior methods that either rely too heavily on attention scores or ignore query conditioning.

The paper communicates technical details clearly, with motivating examples (like flower/pineapple/lime cases) and well-constructed figures that highlight both shortcomings of existing methods and the advantages of Script.

**Audience:**

Yes

**Audience Explanation:**

See above

**Claims And Evidence:**

Yes

**Claims Explanation:**

See above

**Requested Changes:**

N/A

---

> ### Author Response · Authors · 2025-09-03
>
> Dear Reviewer w6Ja,
>
> Thank you for your thoughtful feedback. We truly appreciate the time and care you've taken in reviewing our work. We remain committed to addressing any remaining concerns you may have during the discussion phase and will resolve any issues at the earliest opportunity.
>
> With best regards,
> The Authors of Paper #5413

---

### Review · Reviewer_CQeb · 2025-08-29

**Summary Of Contributions:**

## Summary of the paper

This paper introduces "Script," which is a training-free visual token pruning method designed to mitigate the high computational cost associated with the large number of visual tokens in Multimodal Large Language Models (MLLMs).
The authors identify the key limitations of existing pruning paradigms: attention-based methods are susceptible to biases like "attention sink," while similarity- and divergence-based methods are often query-agnostic, leading to the removal of task-relevant information.

To address this, Script employs a two-main process. (1) Graph-Structured Pruning (GSP) constructs an efficient bipartite graph to model both local and long-range visual redundancy, allowing for the effective removal of semantically repetitive tokens. (2) Query-Conditioned Semantic Pruning (QCSP) leverages the user's text query to score the relevance of the tokens and uses a Determinantal Point Process (DPP) to select a final subset that is not only relevant but also diverse.
A simple intersection-based fusion after the two main processes keeps tokens that are both visually compact (GSP) and query-aligned (QCSP).

## List of strengths and Weaknesses
### 1. strengths
- Strong results and ablations: Script consistently matches or beats FastV, TRIM, VisionZip, DivPrune, and SparseVLM across token budgets; ablations show complementary gains from GSP and QCSP (e.g., 92.19% relative accuracy with only ~2.7% tokens retained).
- The DPP kernel cleanly balances relevance and diversity; the derivation clarifies why det(L_I) discourages redundant selections
- Thorough Analysis and Presentation: The paper is exceptionally well-written and structured. The motivation for the method is clearly established.
### 2. weaknesses
- Modality scope (text–vision only). It seems the author treats MLLM as a vision–language model.
The evaluation setup assumes queries and context are restricted to text and visual tokens; the authors do not model audio (or other modalities such as depth or IMU). Reported efficiency/accuracy gains on vision–language benchmarks therefore may not generalize to audio-conditioned or audio-visual tasks without adding an audio encoder, modality-aware token budgets, and temporally aware redundancy scoring and evaluation.
- Missing sensitivity study for tunables (for example, $\gamma$ in GSP’s redundancy score)
- (Minor) Heuristic for Final Token Selection: The final token set is determined by the intersection of the GSP and QCSP outputs. The paper mentions a fallback mechanism to supplement this set if the intersection is too small, but a deeper analysis of the dynamics of this intersection (e.g., typical overlap size, frequency of fallback) would provide a more complete picture of the interplay between the two modules.

**Audience:**

Yes

**Audience Explanation:**

Token pruning for MLLMs is a timely topic for scalability and deployment. Script’s training-free, query-aware, and diversity-promoting approach, coupled with substantial speedups and KV-cache savings on widely used LLaVA testbeds and standard benchmarks, should interest both efficiency and multimodal reasoning researchers

**Broader Impact Concerns:**

N/A. The authors have included a "Broader Impacts" section (Section G) that adequately addresses the potential societal implications of this work.

**Claims And Evidence:**

Yes

**Claims Explanation:**

This paper presents a strong and well-validated contribution. The authors first clearly articulate the limitations of existing pruning methods, including the "attention sink" phenomenon, which effectively motivates their proposed solution of query-conditioned pruning.

The proposed method Script is also well explained. The authors provide a clear mathematical foundation using Determinantal Point Processes (DPP) and detail an efficient greedy algorithm for implementation. The empirics are equally compelling, demonstrating consistent performance gains over strong baselines across multiple models (LLaVA-1.5, LLaVA-NeXT, Video-LLaVA) and token budgets.

The comprehensive efficiency analysis (reporting FLOPs, latency, KV cache, memory, and accuracy) shows that Script achieves a superior efficiency-accuracy trade-off.

**Requested Changes:**

- Hyperparameter sensitivity & guidance. Provide analysis for $\tau$ (graph threshold), $\gamma$ (Eq. 9 scaling), target $k$ (selected tokens), and kernel choices ($S$ vs. $S’$). Include recommended default values and ranges, and robustness plots across several datasets.
- Video-specific notes. Briefly discuss temporal redundancy handling and whether GSP considers frame-wise vs. spatiotemporal neighborhoods; connect to the Video-LLaVA results table.
- Intersection dynamics (GSP ∩ QCSP). Quantify the interplay between modules: typical overlap size, variance across datasets/prompts, and fallback frequency (how often/top-up amount when $|\text{GSP} \cap \text{QCSP}|<k)$. Add a short table or histogram and discuss when each module dominates.
- Qualitative Failure Case Analysis: The case study in Figure 5 provides an excellent visualization of a success case. To provide a more balanced perspective, it would be valuable to include and discuss a qualitative failure case.
- Modality scope exploration. A short paragraph in Limitations clarifying that the method (and pruning scores) are tuned to text–vision. Outline what would be needed to extend to audio (encoder, modality-aware budgets, temporal/spectral redundancy scores) and why results may or may not directly transfer.

---

> ### Author Response · Authors · 2025-09-03
>
> Dear Reviewer CQeb,
>
> Thank you for your thoughtful and constructive feedback. We truly appreciate the time and care you’ve taken in reviewing our work.
>
> > ### 1.Modality scope exploration.
>
> Thank you for your valuable suggestions about the distinction between VLMs and MLLMs.
>  1) We clarified in Sec.2 and Sec.3.1 that our task specifically targets text–vision question answering.
>  2) The naming convention for MLLM follows prior works like DivPrune[1].
>  3) As shown in Sec.F, extending our work to other modalities is beyond the scope of this study.
>
> > ### 2.Sensitivity study.
>
> Thank you for raising the concern about hyperparameters. We have updated the paper with the analyses in Sec.D.3.
>
> 1) Graph threshold $\tau$: Varying $\tau \in \{0.1,0.3,0.5,0.7,0.9\}$ shows consistent performance; $\tau=0.3$ achieves the best trade-off, as higher thresholds may prune informative tokens.
> 2) Scaling parameter $\gamma$ (Eq.~9): Testing $\gamma \in \{1,5,10,50,100\}$ reveals small fluctuations; $\gamma=5$ provides the most reliable performance.
> 3) Kernel choice: $S'$ consistently outperforms $S$, as it better preserves semantic diversity among retained tokens.
>
> We set $\tau=0.3$, $\gamma=5$, and kernel $S'$ as the default.
>
> Note that $k$ in DPP is not a tunable hyperparameter but directly specifies the number of retained tokens, equivalent to the pruning ratio $p$ in GSP. Detailed results under different $k$ are reported in Sec.5 and Appendix D.
>
> > ### 3.Dynamics of token selection (GSP $\cap$ QCSP).
>
> We analyze the interplay between GSP and QCSP and update in Sec.D.4:
>
> 1) Overlap size. Tab.D.5 shows that the overlap ratio between GSP and QCSP is moderate and decreases with higher pruning ratios. This is expected, since retaining fewer tokens (e.g., 64, 32, or 16) makes agreement harder: GSP emphasizes global redundancy, while QCSP prioritizes query-conditioned relevance.
>
> 2) Fallback frequency. Since the overlap never reaches 100%, the fallback to QCSP is consistently triggered to meet the target number of tokens. As pruning becomes stricter, the fallback contributes a larger fraction of the retained tokens, reflecting QCSP’s growing role in supplementing the final set.
>
> 3) Module contributions. Although QCSP dominates the top-up at high pruning ratios, GSP remains crucial. Ablations confirm that GSP reduces globally redundant tokens, improving efficiency and preserving complementary information that QCSP alone may miss.
>
> The analysis shows that as pruning ratio increases, overlap shrinks, fallback frequency rises, QCSP contributes more tokens, and GSP complements performance, jointly balancing structural pruning with semantic adequacy.
>
> > ### 4.Video-specific notes.
>
> Thank you for the suggestion on spatiotemporal redundancy.
>
> 1) Scope. Our work focuses on visual token pruning. For video experiments (Tab.4), we follow prior pruning works by uniformly sampling frames and applying the same inference framework as in images. This setting already yields competitive results on Video-LLaVA.
>
> 2) Current handling. In this setup, GSP operates frame-wise and does not explicitly model spatiotemporal neighborhoods. Hence, temporal redundancy across frames is not exploited in the current implementation.
>
> 3) Future work. In Sec.F, we discuss that extending GSP to spatiotemporal pruning would require redundancy metrics tailored to temporal correlations. We consider this a promising direction, since temporal redundancy in videos could make pruning even more effective.
>
> > ### 5.Error cases analysis.
>
> To provide a balanced view, we have added a dedicated Sec. E.3 with Fig.E.4, where we show and analyze three representative failures:
>
> 1) Model capability limits. Even when GSP and QCSP preserve query-relevant tokens, the base LLM may still fail to link them to the query, showing that pruning cannot overcome fundamental reasoning bottlenecks.
>
> 2) Text–vision misalignment. Relevance scoring sometimes drifts to visually similar but semantically incorrect regions (e.g., “chair” vs. “person”), revealing weaknesses in cross-modal grounding and showing that pruning is bounded by encoder alignment quality.
>
> 3) Pruning-induced errors. When both GSP and QCSP mis-score tokens, critical evidence (e.g., warning signs) can be pruned. Such compounded errors demonstrate how aggressive pruning itself can create information bottlenecks that downstream reasoning cannot recover from.
>
> These cases also point to concrete directions for improvement, such as stronger cross-modal alignment, adaptive pruning thresholds, and multimodal consistency checks.
>
> We remain committed to addressing any remaining concerns you may have during the discussion phase and will resolve any issues at the earliest opportunity.
>
> With best regards,
>
> The Authors of Paper # 5413
>
> [1] Alvar, Saeed Ranjbar, et al. ``Divprune: Diversity-based visual token pruning for large multimodal models." Proceedings of the Computer Vision and Pattern Recognition Conference. 2025.

---

### Review · Reviewer_uTeB · 2025-09-02

**Summary Of Contributions:**

This paper proposes a graph-based method to identify and remove locally redundant visual patches based on visual similarity, effectively compressing visual tokens while preserving semantic information. Extensive validation demonstrates superior performance and robustness across various large-scale models (including 13B parameter models) and datasets, outperforming prior methods in accuracy and efficiency even after substantial token reduction.

**Additional Comments:**

N/A

**Audience:**

Yes

**Audience Explanation:**

The paper presents a new method for visual token pruning in multimodal large language models, which addresses challenges related to inference efficiency and deployment costs.

**Claims And Evidence:**

Yes

**Claims Explanation:**

The claims made in the submission are supported by accurate, convincing, and clear evidence. The paper provides visualizations, such as relevance maps and token retention visuals, which illustrate how the pruning strategy effectively retains task-relevant visual information. It also demonstrates the robustness and effectiveness of the proposed method through extensive quantitative evaluations across multiple benchmarks and models.

**Requested Changes:**

The proposed approach relies on a threshold. It would be better to offer more detailed guidelines or adaptive strategies for selecting the similarity threshold τ, potentially including an automated or learning-based approach, to enhance reproducibility and ease of use.

Font in Figure 7 can be smaller.

---

> ### Author Response · Authors · 2025-09-03
>
> Dear Reviewer uTeB,
>
> Thank you for your thoughtful and constructive feedback. We truly appreciate the time and care you’ve taken in reviewing our work.
>
> 1. Analysis of threshold $\tau$.
> As shown in Table D.4, our method is highly robust to the choice of $\tau$: across the range $\tau \in [0.1, 0.9]$, the average relative performance consistently exceeds 95\%. The performance also varies smoothly and predictably with $\tau$, without instability or abrupt drops, confirming the stability and reliability of our approach under different settings. Further sensitivity analysis is provided in Section D.3.
>
> We additionally discuss the potential for adaptive or learning-based strategies for selecting $\tau$ in the updated Limitations section (Section F).
>
> 2. Optimization of Figure 7.
>
> Following your suggestion, we have updated Figure 7 to use a smaller, more readable font in the main paper.
>
> We remain committed to addressing any remaining concerns you may have during the discussion phase and will resolve any issues at the earliest opportunity.
>
> With best regards,
>
> The Authors of Paper \# 5413

---

### Review · Reviewer_ok18 · 2025-09-12

**Summary Of Contributions:**

The paper’s subject is in the area of visual token pruning to enhance performance of multimodal large language models (MLLMs) while preserving high accuracy by integrating query relevance queues.

The approach comprises a graph-structured pruning module that removes visually redundant tokens (Thus enhancing performance (Memory and computation time requirements)) and a query-conditioned semantic pruning module that preserves query-relevant visual information contrary to many existing approaches that focus only on eliminating visual redundancy.
The vision-token pruning uses Cosine similarity distance.

**Additional Comments:**

Is formula (2), pg 5 complete?

**Audience:**

Yes

**Audience Explanation:**

Overall, the proposed approach combines query semantic-relevancy with more used visual token pruning, while achieving on-par performance with other existing approaches.

**Broader Impact Concerns:**

NA.

**Claims And Evidence:**

Yes

**Claims Explanation:**

Authors claims a new pruning approach (Script) that requires no retraining and generalizes across diverse MLLMs.
Experiments presented are convincing and use of a bipartite graph as built using the cosine similarity yielding accuracy in par with other approaches while processing only one fraction of tokens hence reducing in a significant proportion, processing power and time needed.

**Requested Changes:**

- Regarding authors’ inferred Insight 1, one can also estimate that similarity of near vs long-range tokens depends on the type of image/scene analyzed and impacts the shape of the right side part of Figure 3(a) Average Cosine similarity vs spatial distance curve.
- Generalization to various types of image data sets has to be carefully selected though as many similar textures/patterns, objects of interest repeat either in indoor images or outdoor scenery. These patterns might pertain to semantically similar objects (furniture, wall, decorations, utensils etc.) in indoor images or patterns/textures in outdoor scenery.
- On the POPE Benchmark with a 90% pruning ratio, the authors’ approach achieves an accuracy of 84.49%, while the random approach is equally fast but drops to 78.63% accuracy.
     - Would this be related to the markedly higher semantic importance/attention scores of certain tokens compared to the majority of the other tokens in the chosen images?
     - Would this difference be lower for more homogeneous token images?

---

> ### Author Response · Authors · 2025-09-13
>
> Dear Reviewer ok18,
>
> Thank you for your thoughtful and constructive feedback. We truly appreciate the time and care you’ve taken in reviewing our work.
>
> > ## 1. Regarding the paper’s inferred Insight 1
>
> (1) We agree that similarity trends may depend on scene types. Our analysis of 10,000 COCO images was intended to provide broad evidence of this phenomenon. While no single dataset can capture all possible scenarios, COCO includes both indoor and outdoor environments with diverse scene complexity, and its diversity makes it a representative testbed for analyzing redundancy patterns. This analysis directly motivated the design of GSP, which considers both nearby and long-range similarity.
>
> (2) In real-world settings, objects within an image may vary in size, yet nearby tokens typically capture similar features because natural images exhibit strong spatial smoothness: adjacent patches are likely to share textures, colors, or edges. Figure~3(a) shows that similarity increases again at larger spatial distances. This effect can be attributed to repeated or periodic patterns (e.g., tiles, fences, grass, sky, ripples) as well as multiple instances of similar objects distributed across the image.
>
> (3) Beyond this analysis, the pruning results further validate Insight 1, showing that explicitly modeling both local and long-range redundancy yields consistent gains and supports the generality of our conclusion beyond COCO.
>
> > ## 2. Generalization to various types of image datasets
>
> We conducted experiments on representative benchmarks and observed consistent results. We believe these findings are relatively generalizable, as pruning addresses redundancy at a fundamental level rather than being tied to a single dataset or task. At the same time, the taxonomy of image types is virtually infinite, making it infeasible to cover every case within one work. Extending this analysis to additional datasets (including more domain-specific or texture-heavy benchmarks) is a valuable direction for future research.
>
>
> > ## 3. On the POPE Benchmark, why does our approach reach 84.49\% accuracy while random pruning drops to 78.63\%?
>
> > **3.1 Does the accuracy gain relate to semantic importance or attention scores?**
>
> We believe the concern refers to Table 1. We would like to clarify that this experiment evaluates pruning under visual similarity; it does not explicitly rely on semantic importance or attention scores. That said, our bipartite GSP encourages diversity and non-redundancy, which indirectly increases the chance of retaining semantically useful tokens.
>
> > **3.2 Would the gap shrink for more homogeneous images?**
>
> The reviewer’s definition of ''homogeneous token images'' is not entirely clear, but we interpret it as images with relatively uniform content, textures, or repeated patterns. In such cases, the performance gap to random pruning may shrink. However, the outcome still depends on the specific query. Even in highly uniform backgrounds with many redundant tokens, GSP remains more robust than random pruning because redundancy-aware matching preserves representative tokens rather than discarding them arbitrarily.
> In addition, our Query-Conditioned Semantic Pruning further enhances this effect by conditioning on the query, emphasizing tokens more relevant to the input and reducing the risk of losing semantically important information.
>
> > ## 4. Additional Comments
>
> Thank you for pointing out the issue with Equation 2. The problem was due to a LaTeX compiling error and has been corrected in the revised version; the formula itself is complete.
>
> Sincerely,
> The Authors of Paper \#5413

---

> > ### Comment · Reviewer_ok18 · 2025-09-25
> > **Thanks for the replies and explanations**
> >
> > I would like to thank the authors for their clear replies and the subsequent corrections they made to the paper in the uploaded revision.

---

### Author Response · Authors · 2025-09-03
**Revised Version Uploaded**

Dear Editor and Reviewers,

Thanks for all reviewer's contrusctive comments, we have accordingly revise our paper. We have uploaded a new revision of the paper with the following changes:

1. A new sensitivity analysis of the hyperparameters $\tau$, $\gamma$, and kernel choice has been added, and the corresponding results are reported in Table D.4.
2. Table D.5 has been introduced to illustrate in detail the interplay between the two proposed modules, GSP and QCSP.
3. Section E.3 has been added to present failure case studies, thereby providing a more balanced view of the proposed approach.
4. We have updated and revised Section F to address the concerns regarding the scope of our research and the adaptability of the hyperparameter $\tau$.

We believe these changes make the paper’s contributions clearer and more solidly grounded. Once again, thank you for helping us improve the manuscript.

Sincerely,
The Authors of Paper # 5413

---

### Decision · Action_Editor_LPFE · 2025-10-17

**Recommendation:** Accept as is

**Audience:**

Yes

**Audience Explanation:**

All four reviewers find this paper interesting and worth sharing with TMLR audience. After considering the paper, review, rebuttal, AC agrees with their judgment. The technique proposed by the authors is simple, well-motivated, and effective.

**Claims And Evidence:**

Yes

**Claims Explanation:**

All four reviewers agree the claims in the paper are well-supported by clear and convincing evidence. After examining the paper and reviews/rebuttal, AC sides with the reviewers' judgment.